# Non-linear archetypal analysis of single-cell RNA-seq data by deep autoencoders

**Yuge Wang**[ID][1], **Hongyu Zhao**[ID][1,2,3]*

**1** Department of Biostatistics, Yale School of Public Health, New Haven, Connecticut, United States of America, **2** Program of Computational Biology and Bioinformatics, Yale University, New Haven, Connecticut, United States of America, **3** Department of Genetics, Yale School of Medicine, New Haven, Connecticut, United States of America

* hongyu.zhao@yale.edu

## Abstract

Advances in single-cell RNA sequencing (scRNA-seq) have led to successes in discovering novel cell types and understanding cellular heterogeneity among complex cell populations through cluster analysis. However, cluster analysis is not able to reveal continuous spectrum of states and underlying gene expression programs (GEPs) shared across cell types. We introduce scAAnet, an autoencoder for single-cell non-linear archetypal analysis, to identify GEPs and infer the relative activity of each GEP across cells. We use a count distribution-based loss term to account for the sparsity and overdispersion of the raw count data and add an archetypal constraint to the loss function of scAAnet. We first show that scAAnet outperforms existing methods for archetypal analysis across different metrics through simulations. We then demonstrate the ability of scAAnet to extract biologically meaningful GEPs using publicly available scRNA-seq datasets including a pancreatic islet dataset, a lung idiopathic pulmonary fibrosis dataset and a prefrontal cortex dataset.

**Data Availability Statement:** The scRNA-seq datasets analyzed in this manuscript are all publicly available. The human pancreatic islet dataset was downloaded from the Gene Expression Omnibus (GEO) under accession code GSM2230757. The

## Author summary

Single-cell RNA sequencing (scRNA-seq) techniques enable the profiling of gene expression at the single-cell level, and thus make it possible to uncover the cellular heterogeneity in a complex cell population which is composed of multiple cell types. Due to the complexity of biological system, different cell types may share underlying gene expression programs (GEPs) at different levels. However, such shared patterns are difficult to study by traditional cluster analysis. Based on the assumption that the expression profile of each cell results from a non-linear combination of multiple GEPs, we develop scAAnet, a deep learning model for non-linear archetypal decomposition of scRNA-seq data. We demonstrate that scAAnet is able to both achieve better decomposition performance in simulated data and identify biologically meaningful GEPs that are either cell-type-specific or disease-enriched in three real scRNA-seq datasets. To help interpret results from scAAnet, we also provide downstream analysis tools for the identification of program-specific marker genes. We expect scAAnet can be applied to explore GEPs shared across cells when scRNA-seq is used to study a complex disease or biological system.

IPF lung data was downloaded from GEO under accession code GSE136831. The prefrontal cortex dataset was downloaded from Synapse (https://www.synapse.org/#!Synapse:syn18485175). A Python implementation of scAAnet can be found on GitHub at https://github.com/AprilYuge/scAAnet. Implementation instructions, related tutorials, and analysis code for generating results in this paper are also available at the same GitHub page.

**Funding:** Y.W. was supported by the Yale World Scholars Program (https://medicine.yale.edu/bbs/training/initiatives/csc/) sponsored by the China Scholarship Council. This research was supported in part by NIH R56 AG074015, R01 GM134005, and P50 CA196530 to H.Z.. The funders had no role in study design, data collection and analysis, decision to publish, or preparation of the manuscript.

**Competing interests:** The authors have declared that no competing interests exist.

This is a *PLOS Computational Biology* Methods paper.

## Introduction

Single-cell RNA sequencing (scRNA-seq) has enabled the profiling of RNA expression in tens of thousands of individual cells and is now regarded as the gold standard for defining cell states in a mixed cell population [1]. The identification of cell states is usually realized by cluster analysis, where cells from a given scRNA-seq dataset are grouped into clusters based on the cell-cell similarity in expression profiles. Many downstream analyses including differential gene expression, gene co-expression network, and pathway enrichment can be performed based on these identified cell clusters. Cluster analysis only assigns each cell to a specific cluster and implicitly assumes that cells from the same cluster are homogeneous and their expression profiles are randomly distributed around a mean expression level. However, recent studies [2,3] have suggested that some cell populations are better described as a continuous spectrum of states rather than discrete cell clusters, but cluster analysis cannot capture continuous changes of cell states among cell populations and possible sharing of GEPs between different cell clusters.

Archetypal analysis [4], a type of unsupervised learning method (mathematical details about archetypal analysis can be found in S1 Text), can be utilized to study the expression profile as a mixture of GEPs and to properly identify those GEPs. Unlike cluster analysis which groups observations into distinct clusters, archetypal analysis finds some extremal points (archetypes) in the multidimensional data. By performing archetypal analysis, a given dataset is assumed to be a superposition of various populations or mechanisms and each observation is represented as a convex combination of archetypes. In the context of scRNA-seq, this is equivalent to decomposing each cell's expression profile into a convex combination of GEPs. Decomposition through archetypal analysis can help us understand the organization of cells, uncover heterogeneity, and study the functional identities within cell populations. All the downstream analyses can be performed based on GEPs similar to those performed on distinct cell clusters. For example, we can identify top genes with high expression levels in each GEP or differentially expressed genes (DEGs) among GEPs. Furthermore, if we have pre-labeled clinically meaningful cell types or disease groups, we can study how the relative activity of GEPs differs among those biological states and identify interesting GEPs relevant to a specific cell type or disease.

The traditional archetypal analysis is linear and can be solved by an alternating constrained least squares algorithm [4]. Since archetypes are 'pure' types in the observed data, their values should be non-negative in the context of scRNA-seq data because the expression levels are non-negative. Linear archetypal analysis decomposes an observed expression profile matrix into the multiplication of two low-rank and non-negative matrices. One matrix is a GEP matrix, representing the bases of a low-dimensional space. The other matrix is a usage matrix, storing the linear coefficients of the bases and representing how much each GEP is used across cells. The main limitation of this procedure results from the assumption that signals can be linearly reconstructed. However, genes work coordinately with each other in a complex way and an observed expression profile is more likely generated non-linearly from the combination of GEPs. Therefore, linear archetypal analysis may not uncover the underlying GEPs and reveal the heterogeneity of the usage of those GEPs within cell populations.

Non-linear archetypal analysis methods have been proposed based on kernelization, such as kernel principal convex hull analysis [5]. However, there is no guarantee that kernel-based

transformation makes data well-approximated by a simplex. A relatively new framework for non-linear archetypal analysis borrows ideas from the field of deep learning. Theoretically, neural networks can closely approximate any linear or non-linear functions [6,7] which makes deep learning-based approaches advantageous over kernelization-based methods. Specifically, we consider using an autoencoder [8], an artificial neural network for unsupervised learning, to perform non-linear archetypal analysis on scRNA-seq data. On the one hand, autoencoders have been utilized to perform non-linear archetypal analysis on images in computer science, such as AAnet [9] and DeepAA [10]. On the other hand, autoencoder-based models including scVI [11] and DCA [12] have been applied to process scRNA-seq data and have gained success in tasks including batch effect removal, denoising, dimensionality reduction, and differential expression analysis. However, the existing methods either are not designed specifically for analyzing scRNA-seq data or do not constrain the distance between archetypes and observed data in the latent space, which is one of the assumptions of archetypal analysis.

To take into consideration both the non-linearity and the count structure in scRNA-seq data, we design a new autoencoder-based method to perform non-linear archetypal analysis called scAAnet. Moreover, we provide a regression-based method to perform DEG test among identified GEPs using results from scAAnet. We will introduce the structure of scAAnet and evaluate its performance against competing methods using simulated datasets. We will also illustrate the ability of scAAnet to identify and interpret GEPs in real scRNA-seq data by applying it to a pancreatic islet dataset [13], a lung idiopathic pulmonary fibrosis (IPF) dataset [14], and a prefrontal cortex dataset composed of patients with varying degree of Alzheimer's disease (AD) pathology [15].

## Results

### Overview of scAAnet

Autoencoders can be used for non-linear decomposition of a dataset. Here, we introduce how scAAnet is designed based on an autoencoder to perform non-linear archetypal analysis of count scRNA-seq data (Fig 1). The structure of scAAnet is mainly composed of two parts, an encoder part and a decoder part. The role of the encoder part is to perform a non-linear decomposition of the data by mapping data from a high-dimensional space to a much lower-dimensional space (latent space). The role of the decoder part is to non-linearly reconstruct the data in the original feature space.

The output of the encoder side, matrix $A$, is a latent representation of the data and each row represents a cell usage vector. Therefore, matrix $A$ should be non-negative with row sum being 1. This constraint can be easily achieved by applying Softmax transformation on the output of the encoder. The other challenge is to make inferred archetypes tight to the data in the latent space. We solve this by setting the positions of archetypes ($Z^{fix}$) in the latent space and penalize the distance between the preset archetypes and inferred archetypes in the latent space. A similar idea was used in DeepAA [10]. Basically, archetypes are set so that the centroid of the simplex formed by these archetypes is the origin in the latent space and the distances between any two adjacent archetypes are the same. Moreover, we make the encoder not only output the coefficient matrix $A$ but also another coefficient matrix $B^T$ and we add an archetypal loss to make sure the archetypes are as close to the convex hull of data as possible in the latent space (Methods and S1 Text section 1).

Usually, mean squared error (MSE) is used as the reconstruction loss for data decomposition methods like archetypal analysis and the underlying assumption is that data values are normally distributed. However, this is not the case for scRNA-seq data which are count data with a large proportion of zeros. To account for the count structure and overdispersion of

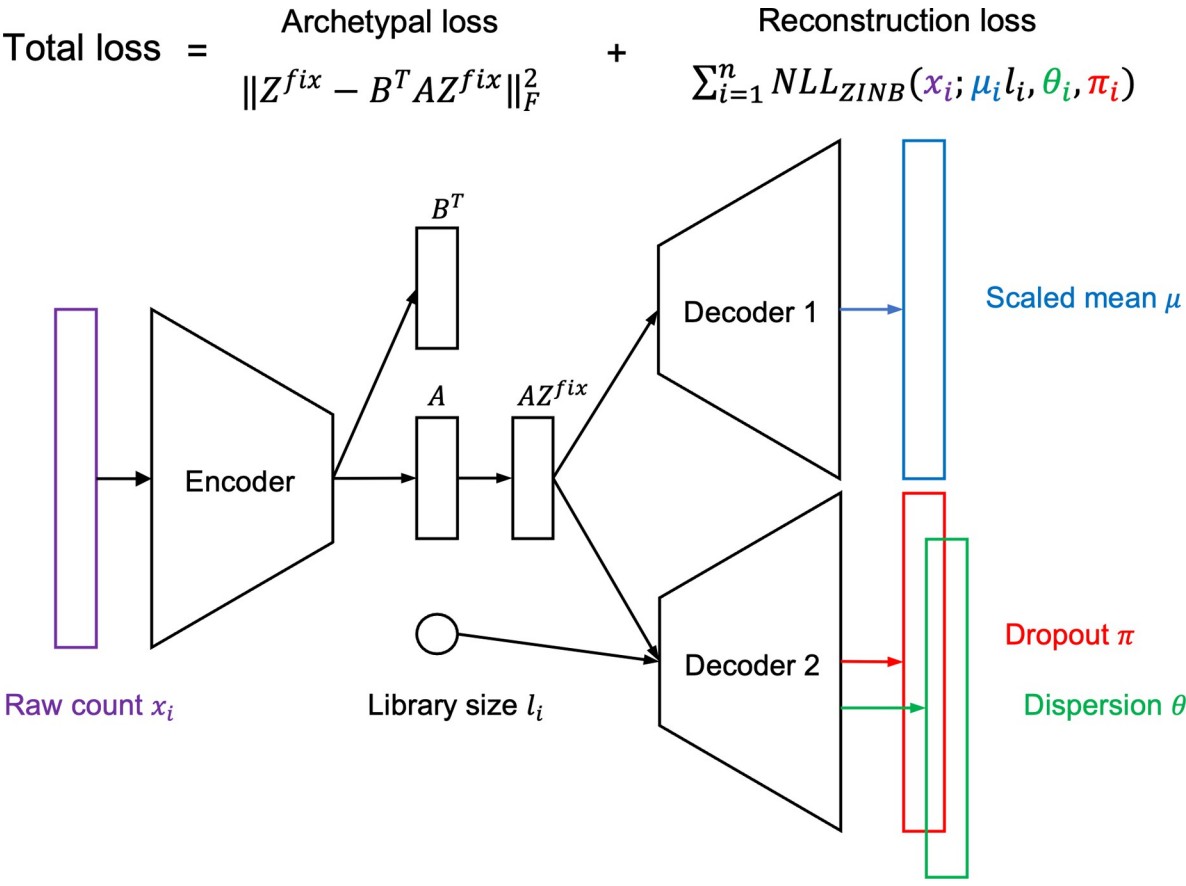

**Fig 1. Structure of scAAnet for non-linear archetypal analysis with a ZINB reconstruction loss.** NLL: Negative log-likelihood.

scRNA-seq, we use a zero-inflated negative binomial (ZINB) distribution to model the observed expression data so that the corresponding reconstruction error is given based on the negative log-likelihood (Methods). What is shown in Fig 1 is the case for ZINB distributions which have three parameter sets: scaled mean $\mu$, dropout probability $\pi$, and dispersion $\theta$. Apart from ZINB, we also provide users with three other distribution choices including Poisson, zero-inflated Poisson (ZIP), and negative binomial (NB). The parameters for these four distributions are shown in Table 1.

The total loss of scAAnet is composed of both the archetypal loss and the reconstruction loss and one can change the relative weight of the two losses. Through simulation studies we found that the performance of scAAnet was robust to the change of the weight around 1, so we set the relative weight of the archetypal loss as 1 by default (see S1 Text section 2 for more discussion).

**Table 1. Parameters for different distributions.**

| Distribution | Parameters |
|---|---|
| Poisson | $\mu$ |
| ZIP | $\mu,\pi$ |
| NB | $\mu,\theta$ |
| ZINB | $\mu,\theta,\pi$ |

There are two matrices we are interested in when applying scAAnet to scRNA-seq data. The first matrix is a usage matrix which contains the coefficients of the archetypal decomposition of all cells. This matrix is the $A$ matrix generated by the encoder. The second one is a GEP (archetype) matrix with each row corresponding to a GEP in the feature space. Values in a GEP vector represent the relative gene expression levels in that GEP. This matrix can be obtained by feeding decoder 1 with $I_K Z^{fix}$ where $I_K$ is a K-dimensional identity matrix.

## Simulation study

We simulated synthetic scRNA-seq data under the framework of Splatter [16] using four archetypes, and ran scAAnet and four competing methods including non-negative matrix factorization (NMF) [17], scVI [11], LDVAE [18] and AAnet [9] to infer both GEPs and cell usage vectors. Here, NMF was used to perform linear archetypal analysis. LDVAE is a linearly decoded version of scVI. Details for competing methods are provided in the Methods. We implemented scAAnet with all the four choices of distribution-based reconstruction loss terms. We used one hidden layer with 128 neurons for each encoder and decoder. The batch size was 64 and 200 epochs were run with early stopping if the training did not improve for 100 epochs. We used $K = 4$ to train scAAnet on simulated data. Note that in real data, it would be impossible for us to know the true number of archetypes and we propose to select $K$ based on the stability of both inferred usage and GEP matrices (Methods). As shown in S1 Fig, we were able to select $K = 4$ for simulated data after considering model stability in general. We also explored the robustness of scAAnet to misspecified $K$ in S1 Text section 3. We used MSE to quantify the accuracy of cell usage inference and reconstruction errors, where reconstruction errors were calculated between the inferred scaled mean and the true scaled mean expression levels. The accuracy of GEP inference was measured by the Pearson correlation between inferred GEP vectors and the truth. Benchmarking results from datasets that were simulated under NB and ZINB distributions are shown in Figs 2 and S1, where $\lambda$ is a parameter used in the simulation process to introduce dropout (Methods). The accuracy of inferred cell usages was higher for scAAnet with count distribution-based error terms, especially when ZINB distributions were used. The worst performance was observed for scAAnet with MSE loss which suggests that MSE is not a good choice for measuring the reconstruction error of count scRNA-seq data. Figures in the second column further show that scAAnet achieves the best performance on the inference of GEPs regardless of distribution used to calculate the error term in the loss function. In S26A Fig, we also observed close resemblance between inferred GEP and true GEP by plotting heatmaps of normalized gene scores using the top 20 DEGs in each GEP. Moreover, we observed better performance of scAAnet relative to the other methods consistently across different signal-to-noise ratio levels. Furthermore, almost all methods had a very low reconstruction MSE with the exception of LDVAE, which had the largest reconstruction errors (last column in Fig 2 and S1 Fig). When the dropout proportion became higher in the datasets that followed ZINB distributions, similar trend was consistently observed (S2 Fig).

To show that the existing models for non-linear archetypal analysis, including scVI, LDVAE and AAnet, do not guarantee that the inferred archetypes are tight to the observed data points, we visualized archetypal space of each method including scAAnet by a fast-interpolation-based multidimensional scaling (MDS) (Figs 3 and S3). Briefly, MDS is performed on archetypes in the original space and then the coordinates of each data point are derived from linear interpolation among coordinates of the archetypes in a low-dimensional MDS space using the coefficients of archetypes learned by each method. Among the four models, the inferred archetypes from scAAnet were the tightest to the data points in the archetypal space,

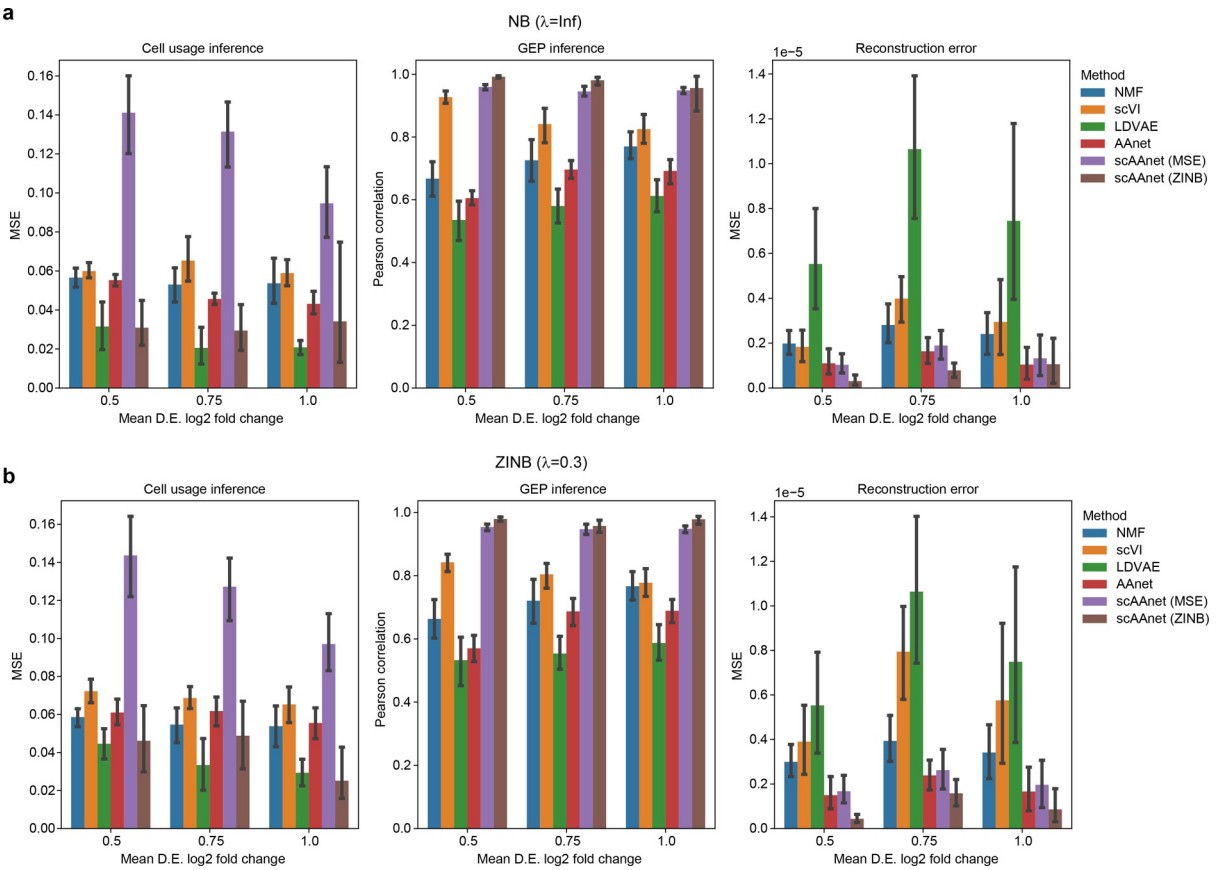

**Fig 2. Performance of scAAnet and other methods on simulated scRNA-seq datasets.** (a) Results from datasets that were simulated under NB distributions. Figures from left to right are MSE between inferred cell usages and true usages of the 4 GEPs, Pearson correlation between inferred GEPs and true GEPs, and MSE between reconstructed scaled means and true means. (b) Similar results from datasets that were simulated under the ZINB distribution ($\lambda = 0.3$). Error bars around the means were drawn with 95% confidence interval. D.E.: differential expression.

where the four archetypes were tightly located at the corners of the data points. However, for the other three models, there was much space between archetypes and data points, especially for AAnet. This indicates that these models do not satisfy the assumption of archetypal analysis that archetypes are extrema in the data space and should resemble the observed data points. Moreover, we can see that for scVI, although the four inferred archetypes were not very far from the data, the inferred usage of GEP 2 and GEP 4 (as shown by the color of true cell usage) did not correspond to any of the four underlying GEPs.

These figures help explain why scVI, LDVAE, and AAnet had relatively good reconstruction but poor decomposition performance. A common goal of these three models is to find a convex hull of a dataset in a transformed low-dimensional space, but there could be multiple convex hulls as long as the data points are included inside the convex hull. Although scVI, LDVAE, and AAnet were designed to find such a convex hull, none of them guarantees that the found convex hull is the closest one to the data. In other words, a good convex hull should be formed by archetypes located near corners of a dataset. Our proposed scAAnet model realizes this by using the archetypal constraint to penalize the distance between archetypes and data points in the latent space.

After getting the decomposition results from scAAnet, an interesting question to ask is to identify GEP-specific marker genes which are upregulated in a GEP compared to others. Since

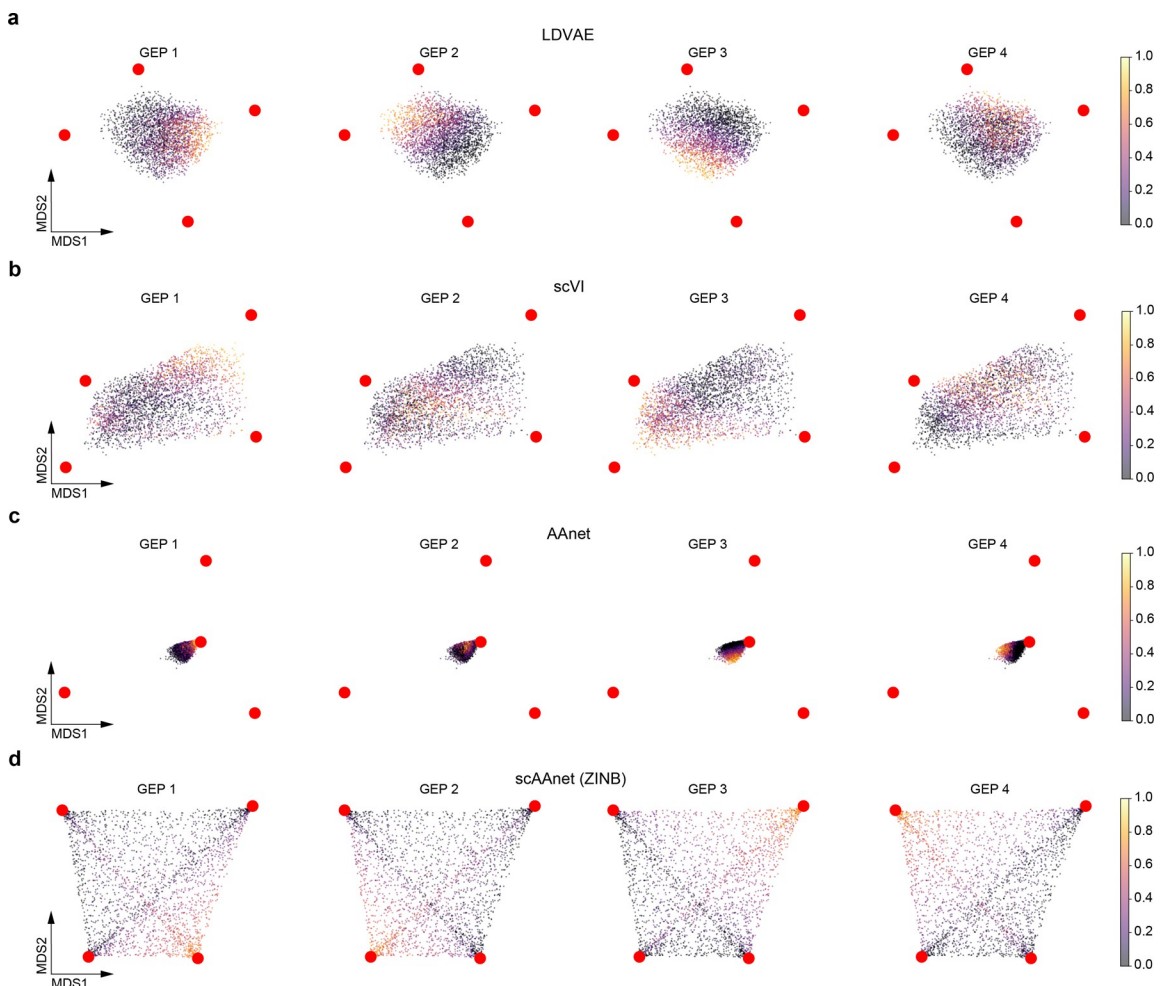

**Fig 3. MDS interpolation visualization of archetypal space.** (a) MDS visualization of archetypal space recovered from LDVAE. (b) MDS visualization of archetypal space recovered from scVI. (c) MDS visualization of archetypal space recovered from AAnet. (d) MDS visualization of archetypal space recovered from scAAnet using ZINB loss. Red dots are the locations of the inferred archetypes. Figures in each column are colored by the true cell usage of the corresponding archetype (GEP). Similar results of scAAnet using other distribution-based loss terms can be found in S2 Fig.

archetypal analysis is unlike cluster analysis where cells are assigned to unique cell clusters, we cannot do two-sample-based tests (e.g., two-sample t test and Wilcoxon rank-sum test) as we do for identifying DEGs among cell clusters. Although we can naively assign each cell to the GEP on which it has the largest usage, it may cause loss of information as the expression profile of each cell can be a mixture of multiple GEPs. Thus, we designed a DEG test accompanying archetypal analysis results from scAAnet based on marginal negative binomial regressions (Methods). Briefly, we use cell usage for each GEP as a continuous variable in the regression model and make the count expression data for each gene as an outcome variable. To show the superiority of using the continuous usage variable to infer GEP-specific DEGs over using the discrete group assignment, we compared the performance of our negative binomial regression model with different predictors using simulated scRNA-seq data.

The median of the area under the curve (AUC) values was consistently higher when we used usage instead of discrete group assignment over different signal-to-noise ratio levels and different distributions (Fig 4). As expected, both methods achieved higher AUC values as the

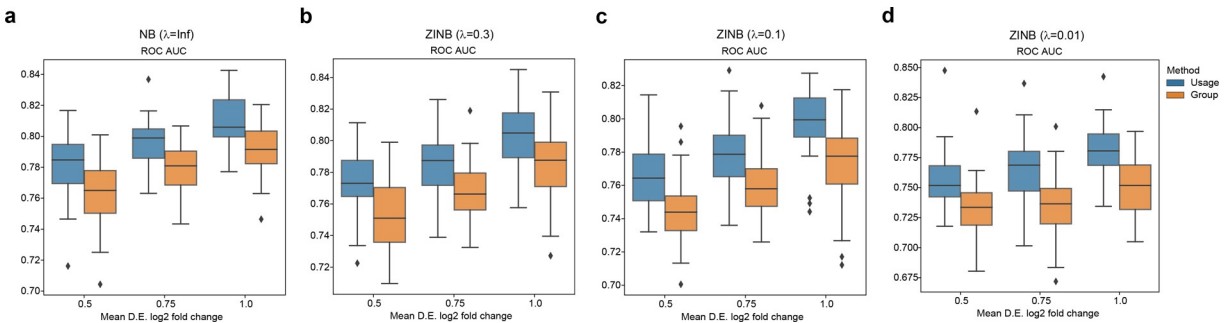

**Fig 4. Comparisons of DEG identifications between using continuous usages and discrete group assignment.** (a) ROC AUC results for datasets simulated under NB distributions. (b) Results for datasets simulated under ZINB distributions with $\lambda = 0.3$. (c) Results for datasets simulated under ZINB distributions with $\lambda = 0.1$. (d) Results for datasets simulated under ZINB distributions with $\lambda = 0.0$. The values of ROC AUC were calculated across different signal-to-noise ratio levels. Each box and whisker plot was plotted based on ten simulated datasets. Central lines represent medians, boxes represent the interquartile range (IQR), and the upper/lower whisker represents the largest/smallest value no further than $1.5 \times$ IQR. ROC: receiver operating characteristic; AUC: area under curve.

signal-to-noise ratio increased because it is easier to identify DEGs when the true mean differential expression fold change was larger. Since adjusted statistical significance level of 0.05 is commonly used as a cutoff to select significant DEGs, we specifically calculated true positive rates (TPRs) and false positive rates (FPRs) for both methods using this cutoff (S4 Fig). Using continuous usage variable resulted in both higher true positive rates and higher false positive rates. This suggests that our model achieved higher sensitivity but relatively lower specificity and our model is less conservative compared to the model using group assignment.

We explored the potential of scAAnet on data simulated purely from one GEP in S1 Text section 4 and on developmental data simulated under linear and bifurcating trajectories in S1 Text section 5.

## Application of scAAnet to a pancreatic islet dataset

Although scAAnet was designed to identify GEPs in heterogenous cell populations without categorically delimited cell state boundaries, we still anticipate that scAAnet is able to capture such discrete cell population structures. We applied scAAnet to a pancreatic islet dataset [13] in which 10 distinct cell clusters were well-defined (Fig 5A). The number of archetypes and the choice of distribution used for computing the reconstruction error were determined by model stability (Methods and S5 Fig). We used outputs from scAAnet with K = 10 and ZINB-based error term as the results of the pancreatic islet dataset. As shown in the two-dimensional MDS plots based on results from scAAnet (S6 Fig), the inferred archetypes were tight to the data in the archetypal space. Since scAAnet belongs to a broader category of decomposition methods, instead of conducting principal component analysis (PCA) and using top PCs to perform Uniform Manifold Approximation and Projection (UMAP), we can use the decomposed usage matrix as a low-dimensional representation of the observed expression profile. The UMAP generated from using the usage matrix (Fig 5A middle) preserved the known cell clusters. Furthermore, we can see similar structure of cell clusters in the UMAP generated from the reconstructed expression profile (Fig 5A right). We also applied NMF to this dataset using K = 10 for comparison. However, results were not satisfactory (see S1 Text section 6 for more discussion). Based on inferred cell usage, most of the 10 GEPs had one-to-one correspondence with annotated cell clusters (Fig 5B–5D). For epsilon cells, the rarest cell type (13 cells) in this dataset, there did not exist any inferred GEPs that uniquely corresponded to them. However, signals from both GEP 3 and GEP 5 were detected in epsilon cells (Fig 5D), indicating the

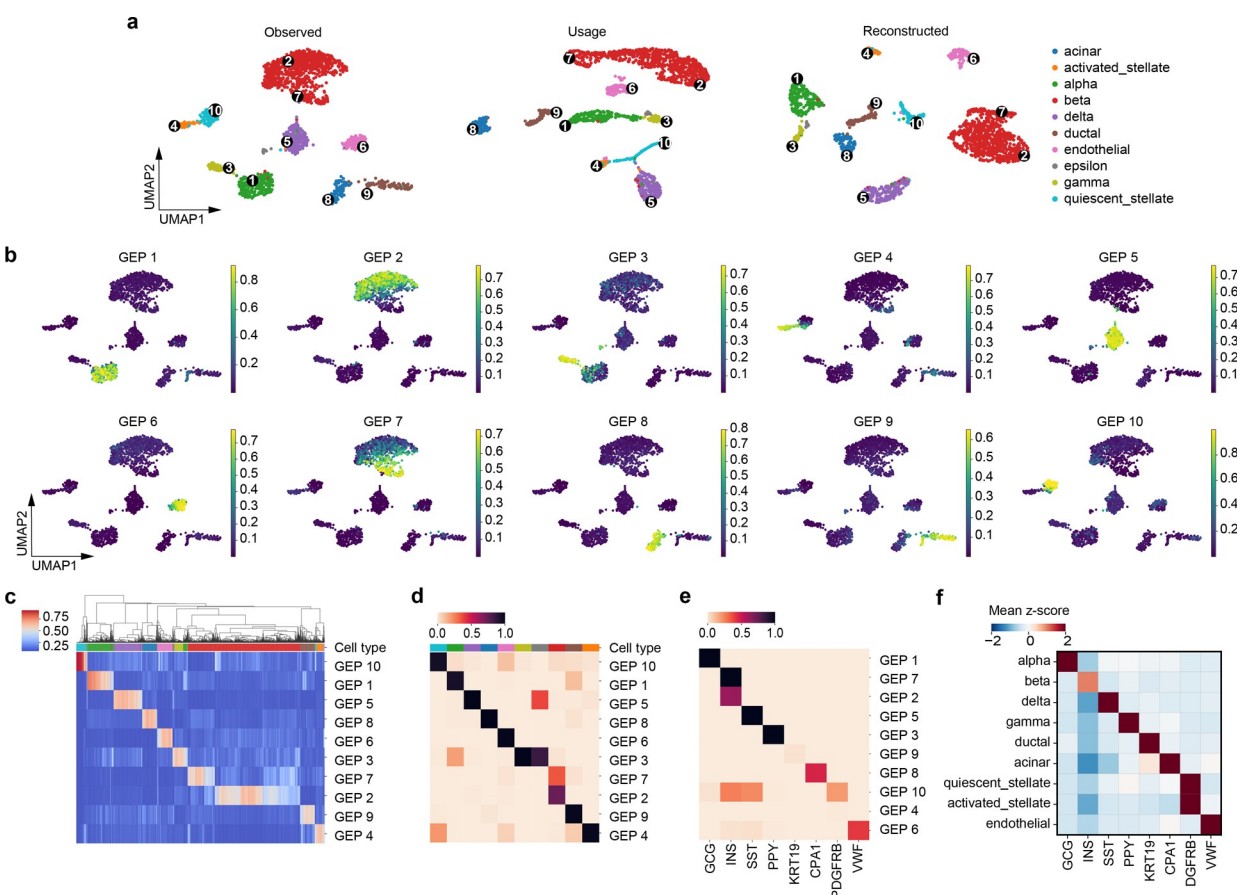

**Fig 5. scAAnet identified archetypes with correspondence to known cell types in the pancreatic islet dataset.** (a) The UMAP visualization using the top 35 PCs of the observed scRNA-seq data (left), using the inferred cell usage matrix from the encoder (middle), and using the reconstructed expression profile from the output layer of scAAnet (right). UMAPs are colored by 10 known cell clusters. The average Silhouette scores for the three UMAPs are 0.604, 0.398 and 0.613, respectively. Black dots are locations of cells that have the largest usage of the corresponding GEP (marked in Arabic numerals). (b) UMAPs from the observed data colored by inferred cell usage for each GEP. (c) Heatmap showing the usage of all GEPs (rows) in all cells (columns). Cells are ordered by hierarchical clustering. (d) Heatmap showing the percentage of cells with usage > 25% of each GEP (rows) in each cell type (columns). Colors in c and d are coded in the same way as those in a. (e) Normalized gene scores (Methods) of known markers (columns) in each GEP (rows). (f) Mean z-scores of known markers (columns) in each cell cluster (rows).

similarity between epsilon cells and gamma cells as well as delta cells, respectively. Note that these cell types are all endocrine cell types in pancreas.

As the correspondence between GEPs and cell clusters was established based on cell usage, we further looked into the inferred GEP matrix to see if known marker genes had higher gene scores in the corresponding GEP. Marker genes were selected based on published papers and were validated in this dataset (Fig 5F). For example, GCG, INS, PPY, and SST are endocrine markers that are highly expressed in alpha, beta, gamma, and delta cells, respectively [19,20]. KRT19 and CPA1 are exocrine markers that are expressed in ductal and acinar cells, respectively [21]. While PDGFRB is a marker of stellate cells [22] and VWF a marker of endothelial cells [23]. The normalized gene score of each selected marker gene was high in the GEP that had correspondence with the cell type that it was linked to (Fig 5E), confirming the consistency between annotated cell clusters and inferred GEPs. We also identified top genes for each GEP by fitting a Gamma distribution to the normalized gene scores (Methods). The top genes for

each GEP are listed in S1 Table and the prementioned marker genes all showed up in the top gene list of the corresponding cell types.

Among all the cell types, the population of beta cells was the only one that was mapped onto more than one GEP, including GEP 2 and GEP 7 (Fig 5B–5D). To uncover the biological characteristics of the two GEPs, we performed DEG tests on beta cells for GEP 2 and GEP 7 separately. We identified 26 and 84 significantly upregulated genes specific to GEP 2 (S2 Table) and GEP 7 (S3 Table), respectively (Bonferroni adjusted p-value < 0.05). The top 20 upregulated DEGs (normalized gene scores in S26B Fig) with the smallest adjusted p-values are plotted in Fig 6B and 6D in which cells are ordered by the inferred cell usage. The expression pattern of those identified DEGs covaried with the inferred cell usage. For GEP 2, we found that several genes among the top 20 upregulated DEGs are functional beta genes (e.g., IAPP, CHGA, MAFA, and ABCC8). For GEP 7, some identified top genes are related to endoplasmic reticulum (ER) stress, including HERPUD1, HSPA5, DDIT3, and PPP1R15A [24].

Moreover, we performed gene set enrichment analysis on all upregulated DEGs of GEP 2 and GEP 7 separately. Enriched Gene Ontology (GO) terms for GEP 2 are related to functions of beta cells (Fig 6A and S4 Table), including peptide hormone secretion and insulin secretion. This result is consistent with the finding that many functional beta genes were identified in the top upregulated genes of GEP 2. GEP 7 is most significantly enriched for gene sets related to stress response (Fig 6C and S5 Table), among which the top three are response to ER stress, cellular response to topologically incorrect protein, and stress response to metal ion. We also conducted enrichment analysis using the Reactome database (S7 Fig, S6 and S7 Tables). The top three terms enriched for GEP 2 are gluconeogenesis, glucose metabolism, and regulation of gene expression in beta cells. For GEP 7, terms similar to those identified in the GO enrichment results showed up in the top terms of Reactome results, including cellular response to external stimuli and Unfolded Protein Response. Results from Reactome further confirmed the biological features of GEP 2 and GEP 7.

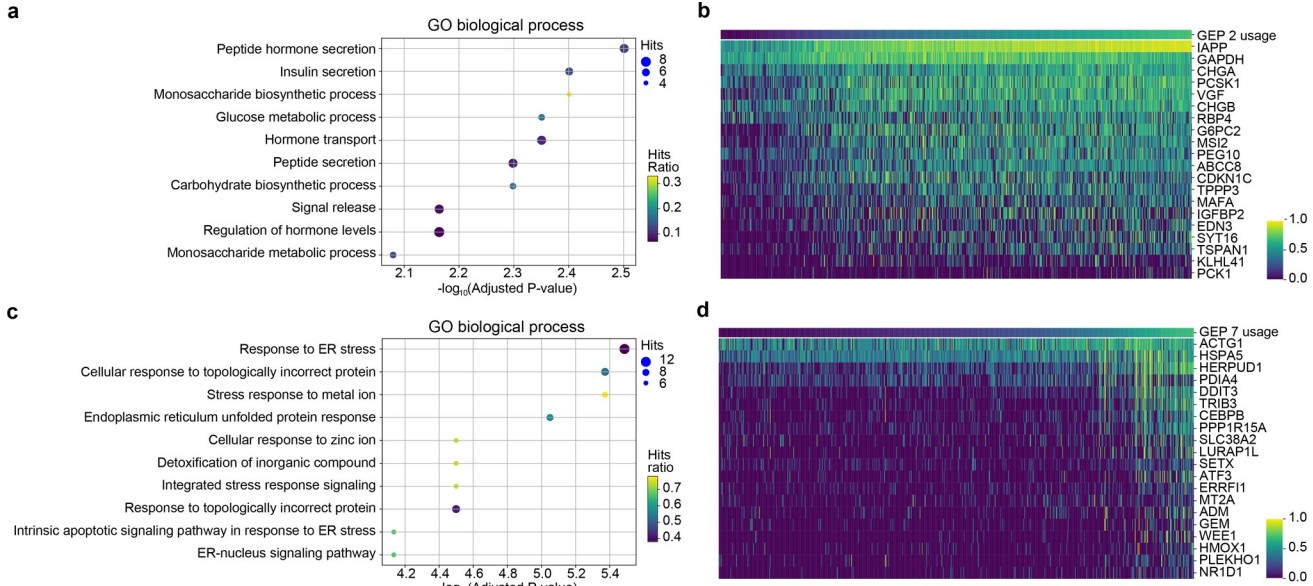

**Fig 6. Enriched GO terms and top DEGs of GEP 2 and GEP 7 in the pancreatic islet dataset.** (a) The top 10 enriched GO biological process terms using 26 significantly upregulated genes of GEP 2. (b) The expression levels of the top 20 DEGs of GEP 2 with the largest z-scores from the DEG test covaried with the inferred usage of GEP 2. (c) The top 10 enriched GO biological process terms using 84 significantly upregulated genes of GEP 7. (d) The top 20 DEGs of GEP 7 with the largest z-scores. Cells in b and d were sorted in increasing order based on the inferred cell usage of GEP 2 and GEP 7, respectively.

To show the robustness of scAAnet, we further applied it to beta cells only and identified two GEPs (S8A and S8B Fig). We found high correlation (03C1 $\rho$ = 0.98) between the first GEP and GEP 2 identified using the entire data, and high correlation ($\rho$ = 0.99) between the second GEP and GEP 7 identified using the entire data (S8C Fig). We also performed DEG tests for the two GEPs and identified 35 and 130 upregulated DEGs, respectively (S8 and S9 Tables). The 35 DEGs covered 25 out of 26 DEGs identified in GEP 2 and the 130 DEGs covered 82 out of 84 DEGs in GEP 7. Moreover, the top enriched GO terms of the two GEPs showed that one of them is responsible for the function of beta cells as GEP 2, while the other is involved in stress response as GEP 7 (S8D and S8E Fig).

## Application of scAAnet to an IPF Lung Dataset

IPF is a chronic and progressive lung disease characterized with scarring in distal lungs. Patients with IPF often suffer from shortness of breath and a persistent dry cough [25]. Although the scale of IPF studies is increasing, the cellular and molecular determinants of IPF remain largely unknown. The accumulation of myofibroblasts in lungs of IPF patients has been observed, where myofibroblasts are involved in the inflammatory response to injury by secreting cytokines [26,27]. It is known that myofibroblasts are differentiated from fibroblasts, which synthesize the extracellular matrix and collagen. Therefore, it is interesting to study this specific cell population in IPF lungs. We applied our model to a lung scRNA-seq dataset [14] by specifically selecting myofibroblasts and fibroblasts. The dataset contained cells from 32 IPF, 18 chronic obstructive pulmonary disease (COPD), and 28 control lungs. Again, the number of archetypes and the choice of distribution used for the loss term were determined by model stability. Results with K = 3 and NB distributions were the most stable and achieved relatively low reconstruction error (S9 Fig), so we will present results obtained under this setting.

Fibroblasts and myofibroblasts are next to each without any gaps in the UMAP (Fig 7A and 7B) which confirms that these two cell types are biologically related and this cell population is suitable for archetypal analysis. The three inferred GEPs are located near the corners of the UMAP and this is consistent with the assumption of archetypal analysis that archetypes are extrema in a dataset. Among the three GEPs, GEP 1 is a fibroblast-specific GEP, while GEP 2 and 3 are myofibroblast-specific (Fig 7C–7E and S10 Fig). GEP 3 is the one not only cell-type-specific but also disease-related. Among fibroblasts, the inferred usage of GEP 3 in IPF is significantly higher than that in both COPD (adjusted p-value = 1.79×10$^{-15}$) and Controls (adjusted p-value = 7.13×10$^{-41}$) (Fig 7F), while the difference between COPD and Controls is not significant for GEP 3 (adjusted p-value = 1.00). We identified 21, 47, and 16 top genes for GEP 1, 2, and 3, respectively (S10 Table), based on the normalized gene scores. Among the 21 top genes in GEP 1, there are some genes known to express highly in fibroblasts, such as PLA2G2A, MFAP5, and PI16 [14,28]. In the meantime, SCN7A, NEBL and ITGA8 showed up in the 47 top genes for GEP 2, which are marker genes for myofibroblasts [29]. This suggests GEP 1 and GEP 2 are programs that are typical in fibroblasts and myofibroblasts, respectively. In contrast, GEP 3 is more likely to be a program specific to IPF patients in myofibroblasts.

To further investigate the differences among the three GEPs, especially for GEP 3, we conducted DEG tests for the three programs separately using marginal negative binomial regressions. We focused on the upregulated DEGs whose expression levels were positively related with the inferred cell usage of each GEP (Figs 8B, 8D, 8F, and S26C). The number of upregulated DEGs in GEP 1, 2, and 3 were 234, 494, and 287, respectively (S11–S13 Tables). Among the top 20 upregulated DEGs in GEP 3, we observed markers of epithelial mesenchymal transitions such as COL1A1, COL1A3, FN1, and TNC [30]. Then, we used these upregulated DEGs to conduct gene set enrichment analysis for each GEP separately. Analysis on GEP 1 returned 137 enriched GO

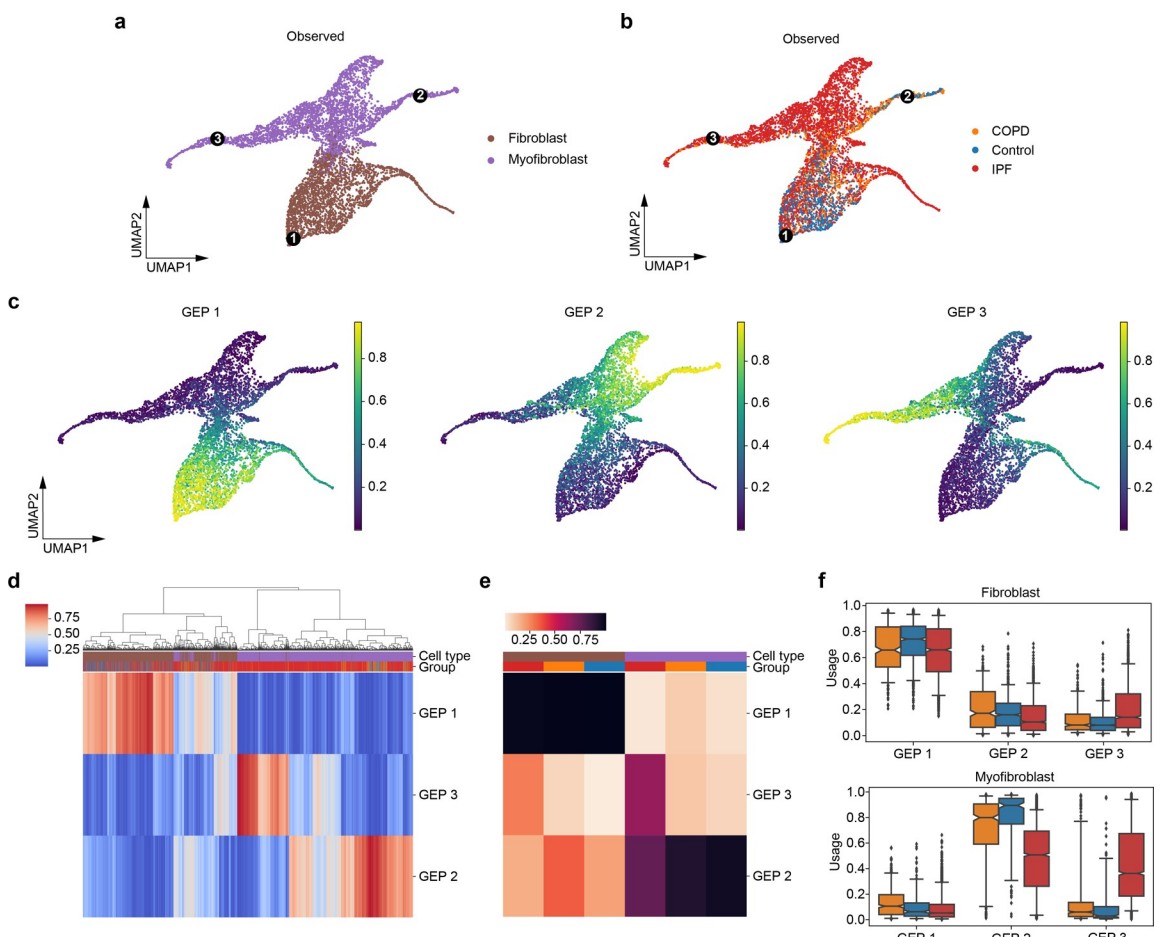

**Fig 7. scAAnet identified 3 GEPs in the lung fibroblast and myofibroblast cells.** (a) The UMAP visualization (Methods) of observed scRNA-seq data colored by cell types. (b) The UMAP visualization of observed scRNA-seq data colored by disease groups. Black dots in a and b are locations of cells that have the largest usage of the corresponding GEP (marked in Arabic numerals). (c) UMAPs colored by inferred cell usage for each GEP. (d) Heatmap showing the usage of all GEPs (rows) in all cells (columns). Cells are ordered by hierarchical clustering. (e) Heatmap showing the percentage of cells with usage > 25% of each GEP (rows) in each cell type and disease group (columns). Colors of cell types and disease groups in d and e are coded in the same way as colors in a and b. (f) Box and whisker plot of the usage of each GEP in cells of fibroblast (top) and myofibroblast (bottom), colored by disease groups (colors are coded in the same way as in b). Central lines represent medians, boxes represent the IQR, and whiskers represent the 5th and 95th quantiles.

biological processes (Fig 8A and S14 Table) among which the top one with the smallest p-value is external encapsulation structure organization (p-value = $4.16 \times 10^{-7}$ and adjusted p-value = $4.17 \times 10^{-4}$). Enriched GO terms for GEP 2 contained many that are related to cell adhesion, migration and tube development (Fig 8B and S15 Table). Part of the enriched terms in GEP 3 are similar to those identified in GEP 2, while we also identified biological processes related to external encapsulating structure organization, connective tissue development, and collagen fibril organization (Fig 8E and S16 Table). Apart from using the GO biological process database, we conducted enrichment analysis using the Reactome database (S17–S19 Tables). Among the top enriched terms, similar pathways were identified which confirmed our results (S11 Fig).

## Application of scAAnet to microglia in a prefrontal cortex dataset

AD is a slowly progressing and incurable neurodegenerative disorder affecting millions of people around the world, but little is known about the molecular complexity of AD [31]. Microglia

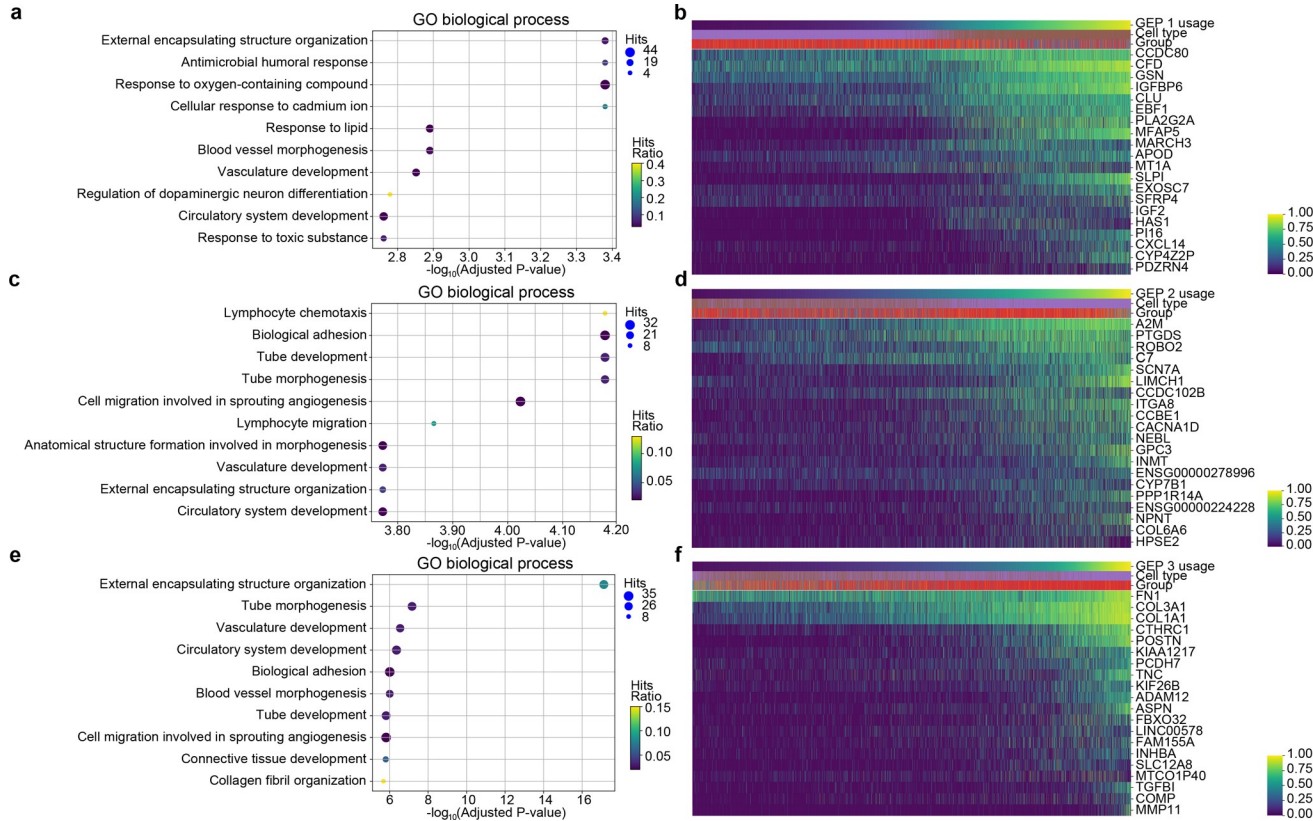

**Fig 8. Enriched GO terms and top DEGs of GEPs in the lung fibroblast and myofibroblast cells.** (a) The top 10 enriched GO terms using upregulated DEGs in GEP 1. (b) The expression levels of the top 20 DEGs of GEP 1 covaried with the inferred usage of GEP 1. (c) The top 10 enriched GO terms using upregulated DEGs in GEP 2. (d) The expression levels of the top 20 DEGs of GEP 2 covaried with the inferred usage of GEP 2. (e) The op 10 enriched GO terms using upregulated DEGs in GEP 3. (f) The expression levels of the top 20 DEGs of GEP 3 covaried with the inferred usage of GEP 3. Top 20 genes in b, d and f were selected based on their z-scores. Cells in b, d and f were sorted by increasing order of the inferred cell usage of GEP 1, GEP 2 and GEP 3, respectively. Colors of cell types and disease groups in b, d and f are coded in the same way as colors in Fig 7a and 7b, respectively.

are the intrinsic macrophages of the central nervous system and have been found to be involved in AD pathogenesis through neuroinflammation [32]. Mathy et al. [15] published the first single-cell-level study of AD pathology in 2019. The study analyzed 48 prefrontal cortex brain specimens selected from the Religious Orders Study and Memory and Aging Project (ROSMAP), among which 24 individuals were defined as AD-pathology and 24 as no-pathology based on the levels of $\beta$-amyloid (A$\beta$) and other pathological hallmarks. The paper discovered four subclusters within microglia, among which Mic1 was found to be both AD-pathology and female enriched (Fig 9A–9C). We applied scAAnet with Poisson distribution-based reconstruction loss to 1,920 microglial cells in this dataset and identified four GEPs that had close correspondence with the four subclusters (Fig 9D). From MDS visualization (S12 Fig), we can observe that each microglia subcluster was enriched at the corresponding corner of the simplex. We performed DEG tests for each GEP (S20–S23 Tables, normalized gene scores in S26D Fig) and the correspondence was further confirmed by the enrichment analysis between the DEGs of each GEP across sets of marker genes of the four microglia subclusters (Fig 9E and S24 Table), which were provided by the original paper. As we can tell, GEP 1, 2, 3, and 4 are mostly enriched in Mic0, Mic3, Mic2, and Mic1, respectively.

Then, we investigated whether the overall usage of the four GEPs differs between AD-pathology and no-pathology group (Figs 9f and S13a), and between females and males in each

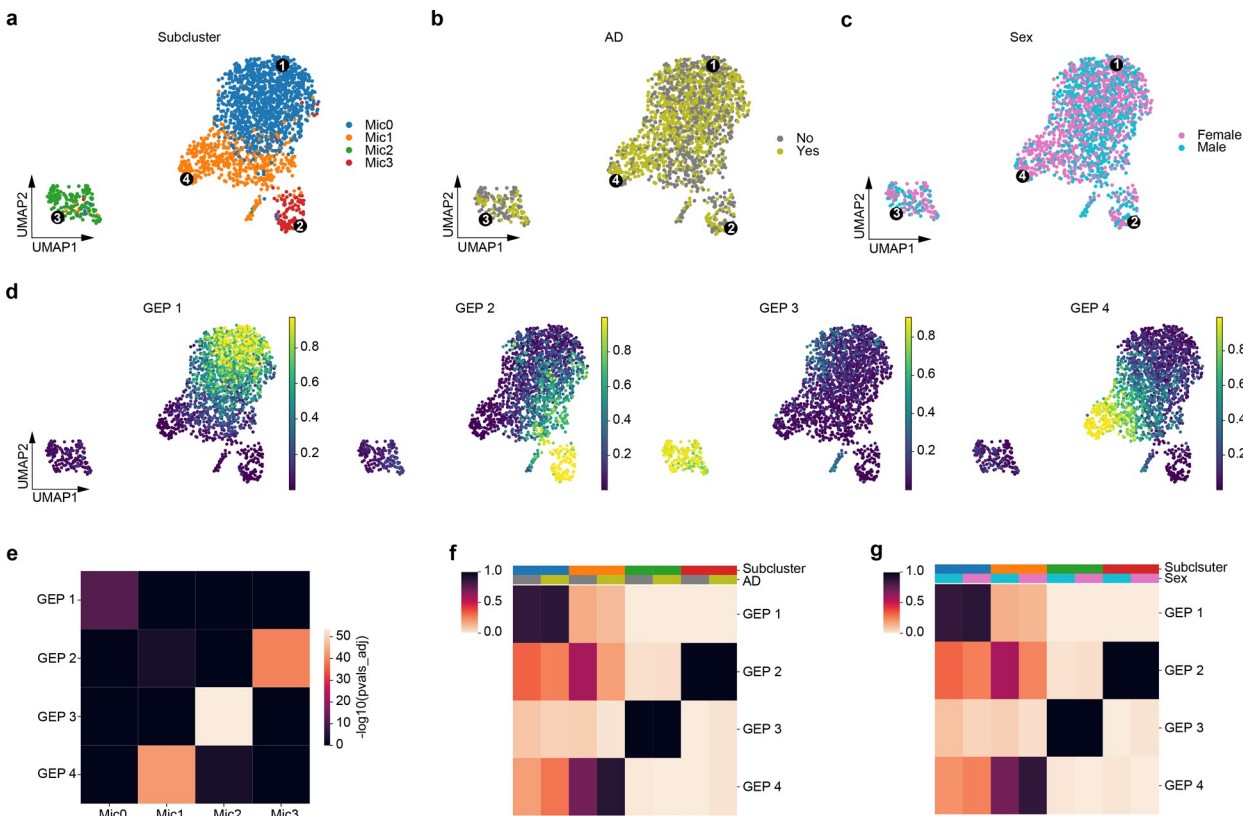

**Fig 9. scAAnet identified 4 GEPs using microglia in the prefrontal cortex dataset.** (a) UMAP of the observed scRNA-seq data colored by microglia subclusters. (b) UMAP of the observed scRNA-seq data colored by AD pathology group. (c) UMAP of the observed scRNA-seq data colored by sex. Microglia subclusters, AD pathology group and sex are provided by the original paper. Black dots in a, b and c are locations of cells that have the largest usage of the corresponding GEP (marked in Arabic numerals). (d) UMAPs colored by inferred cell usage for each GEP. (e) Heatmap showing the enrichment of identified DEGs in each GEP across sets of marker genes of the four microglia subclusters. Colors are negative log of adjusted p-values from hypergeometric tests. P-values were adjusted by Bonferroni over all GEPs and subclusters. (f) Heatmap showing the percentage of cells with usage > 25% of each GEP (rows) in each subcluster and AD group (columns). (g) Heatmap showing the percentage of cells with usage > 25% of each GEP (rows) in each subcluster and sex (columns). Colors of subclusters, AD group and sex in f and g are coded in the same way as colors in a, b and c.

microglia subcluster (Figs 9g and S13b). Among all the subclusters, we observed clear distinction between the two AD groups or the two sex groups in the percentage of cells that used GEP 2 and GEP 4 in Mic1 (Fig 9F and 9G), the only subcluster in microglia that was AD-pathology-associated [15]. Since GEP 4 was found to be mostly enriched in Mic1, it was not surprising that there were more cells using GEP 4 in the AD-pathology or the female group than in the non-pathology or the male group. Moreover, we found that about 58.8% of Mic1 cells in the non-pathology group used GEP 2, while the percentage was only 19.6% in the AD-pathology group. In the meantime, about 58.2% of Mic1 cells from males used GEP 2, while only 25.4% of Mic1 cells from females used GEP 2.

To study the biological functions underlying each GEP, we conducted gene set enrichment analysis using significantly upregulated DEGs in each GEP and identified 65, 108, and 86 enriched GO terms for GEP 2, 3, and 4, respectively (S25–S27 Tables). GEP 2 was found to be involved in biological processes that are related to regulation of phosphorylation and protein modification (Fig 10A and 10B), while GEP 4 was characterized by protein targeting to membrane, RNA catabolic process, and protein localization to ER (Fig 10C). Furthermore, we found many ribosomal protein coding genes (RPS6, RPS11, RPS15, RPS19, RPS20, RPS24,

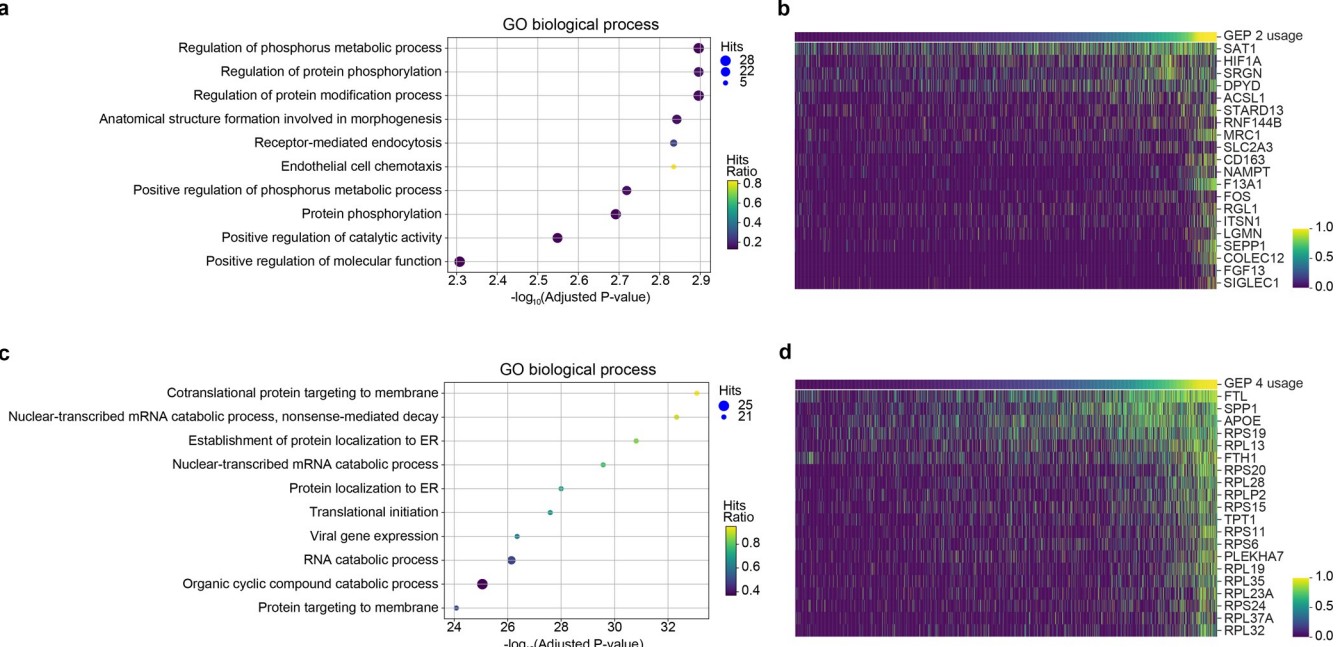

**Fig 10. Enriched GO terms and top DEGs of GEP 2 and GEP 4 in the prefrontal cortex dataset.** (a) The top 10 enriched GO biological process terms using upregulated DEGs in GEP 2. (b) The expression levels of the top 20 DEGs with the largest z-scores covaried with the inferred usage of GEP 2. (c) The top 10 enriched GO biological process terms using upregulated DEGs in GEP 4. (d) The expression levels of the top 20 DEGs of GEP4 covaried with the inferred usage of GEP 4. Cells in b and d were sorted by increasing order of the inferred cell usage of GEP 2 and GEP 4, respectively.

RPL13, RPL19, RPL23A, RPL28, RPL32, PRL35, PRL37A, and RPLP2) among the top 20 upregulated DEGs in GEP 4 ([Fig 10D]). This is consistent with the finding that ribosomal dysfunction might be a crucial event in AD [33]. Besides, APOE was the gene with the third largest z-score among all the DEGs in GEP 4 and APOE can influence the onset of Aβ deposition [34,35], which indicates the importance of GEP 4 in AD pathology.

## Discussion

Traditional cluster analysis on scRNA-seq data groups mixed cell populations into discrete cell clusters, which does not consider the continuous changes of cell states and possible sharing of GEPs among those cell clusters (see S1 Text section 7 for discussion on the level of sharing in the three real datasets). Therefore, archetypal analysis can be a more effective way to model the observed expression profile as a convex combination of GEPs, as this allows the utilization of more than one GEP in a single cell and the sharing of the same GEP among cells [3]. PCA is another commonly used data decomposition method for dimensionality reduction of scRNA-seq data [36]. The difference between archetypal analysis and PCA comes from the non-negativity of the two low-rank matrices and the convexity. The convexity ensures that elements in the usage/coefficient vector of a cell are non-negative and sum to 1. Those elements can be interpreted as how much a cell 'uses' different GEPs or the relative activity of GEPs in a cell. Moreover, the inferred archetypes are some 'pure' types or extrema of the real data, while principal components are not required to resemble 'pure' types in the data and their elements can be either positive or negative. Therefore, archetypal analysis adds to more interpretability than PCA and suits the mixture property of expression data better than cluster analysis.

Here, we present scAAnet, an autoencoder-based model for non-linear archetypal analysis designed specifically for scRNA-seq data. There are mainly three advantages of scAAnet over

the existing methods for archetypal analysis. First, scAAnet is able to perform non-linear decomposition in the case where data are not generated linearly, such as features in high-dimensional biological systems. Second, we measure the reconstruction error by modeling scRNA-seq data using count distributions. We showed that count distribution-based loss led to more accurate results than MSE using synthetic scRNA-seq data. Third, we added an archetypal constraint in the loss function to decrease the distance between archetypes and data points in the latent space. We compared scAAnet against some existing methods that can perform archetypal analysis and showed that it is important to impose the archetypal constraint to find the tightest GEPs. It is worth mentioning that if we replace all non-linear activation functions by identity functions and remove the hidden layer in all decoders/encoders, scAAnet would perform linear archetypal decomposition on count data. This is equivalent to one group of methods based on topic models [37,38], such as latent semantic indexing [39] and latent Dirichlet allocation [40].

We applied scAAnet to three real datasets to showcase the ability of scAAnet to identify biologically meaningful GEPs and illustrate how results from scAAnet can be used for downstream analysis such as identifying program-specific DEGs and gene set enrichment analysis. The first dataset is a pancreatic islet dataset [13] where 10 cell types were clearly defined and categorically delimited in UMAP visualization. We found a good correspondence between GEPs identified by scAAnet and the 10 well studied cell types. Besides, beta cells were related to two distinct GEPs, among which one was responsible for beta cell functions and the other was linked to ER stress response. The results are consistent with findings in published papers [13,41]. However, in the original paper, the authors detected these two 'subtypes' through a more exploratory type of analyses. More specifically, they performed PCA on beta cells and found genes positively correlated with the value of PC1 were functional beta genes and genes negatively correlated with the value of PC1 were involved in the stress response. We think the benefit of our method is that it provides a more systematic and principled way of identifying the heterogeneity within a cell type. The second dataset is an IPF lung dataset. We identified three GEPs using fibroblasts and myofibroblasts and showed that GEP 3 was an IPF-related archetype with higher signals in myofibroblasts. GEP 3 was not only related to cell migration and adhesion, but also involved in collagen fibril and extracellular matrix organization, suggesting its potential role in pulmonary fibrosis [42,43]. Note that in the original paper of this dataset, two pseudotime-based archetypes were identified within myofibroblasts and fibroblasts separately, and myofibroblasts ordered towards the end of the trajectory were found to be enriched in IPF lungs [14]. Therefore, scAAnet was not only able to recover the trajectory-based archetype that was both IPF and myofibroblast enriched, but also identify one more archetype in fibroblasts, suggesting the potential of scAAnet on trajectory-based analysis at a higher resolution. The third dataset is a prefrontal cortex dataset from patients with varying degree of AD pathology. We identified four GEPs using microglia and illustrated the close connection between them and the four microglia subclusters presented in the original paper [15]. Our approach showed that Mic1's association with AD pathology resulted from both the higher activity level of GEP 4 and the lower activity level of GEP 2, demonstrating the ability of scAAnet to reveal cellular heterogeneity at a higher resolution.

There are some limitations and practical issues of scAAnet. First, although scAAnet is able to deliver similar results to cluster analysis on a cell population which contain categorically delimited cell types, we still recommend not to apply scAAnet directly to such a scRNA-seq dataset. This is because scAAnet is a deep learning-based method that involves stochastic training and many hyperparameters and it will take more resources to run compared with traditional cluster analysis. Instead, we recommend users to run scAAnet on a subset of cell types that are likely to involve shared programs (UMAP visualization and pre-clustering combined

with domain knowledge can be used to determine this) or are of particular interest to researchers. Second, the number of archetypes (K) for scAAnet is provided by users and since archetypal analysis is unsupervised, there is no universally definitive criteria to choose the value of K. In practice, we run scAAnet with a set of different Ks and select the one that is reasonably stable with relatively low reconstruction error (Methods). Then, we can vary K around the choice and to see if approximately the same core set of GEPs are identified. Third, the current version of scAAnet is designed for identifying GEPs using single-cell transcriptomic data, but with the recent advances in single-cell multi-omics sequencing technologies [44–48], more simultaneously measured multi-omics data from single cells will be available and scAAnet can be extended to infer GEPs based on multiple modalities jointly. Fourth, although we have shown the potential of scAAnet on trajectory analysis when the topology is simple (e.g. linear or bifurcation), scAAnet cannot replace state-of-the-art trajectory inference methods because most developmental data contain a more complex topology (e.g. multifurcation or circular) [49]. Encompassing such data within a simplex would result in loss of resolution. Therefore, we recommend using scAAnet as a complement for trajectory analysis when the topology is simple, or data are measured within a relatively short developmental period.

Altogether, scAAnet enables the non-linear decomposition of scRNA-seq data and identification of expression programs in mixed cell populations, especially when cell types are related and are not categorically delimited. The latent representation (the usage matrix) of scRNA-seq data can be used as an effective baseline for various visualization tools such as tSNE [50,51] and UMAP [52,53]. We also provide accompanying downstream analysis tools such as top gene identification and DEG tests to help interpret the results from scAAnet. We expect this method can be used as an alternative to cluster analysis in analyzing scRNA-seq data when researchers want to explore shared programs among different cell states.

## Methods

### Design of scAAnet

<u>Architecture and model formulation</u>: scAAnet models count scRNA-seq data with ZINB distributions, so the outputs include not only the mean $\mu$, but also the dispersion $\theta$ and dropout probability $\pi$. We present the architecture of scAAnet in the following specifications for cell $i$:

$$a_i = f_{E_a}(\bar{x}_i)$$

$$b_i = f_{E_b}(\bar{x}_i)$$

$$e_i = a_i Z^{fix} \tag{1}$$

$$\mu_i = f_{D_\mu}(e_i)$$

$$\theta_i = f_{D_\theta}(e_i, l_i)$$

$$\pi_i = f_{D_\pi}(e_i, l_i)$$

where $f_{E_a}$ and $f_{E_b}$ represent two non-linear transformations through the encoder side, and $f_{D_\mu}$, $f_{D_\theta}$, and $f_{D_\pi}$ are decoder neural networks that map the latent representations to the mean, the dispersion, and the dropout probability parameter, respectively. Encoder $E_a$ and $E_b$ are designed to share the same set of layers except for the last one. Similarly, decoder $D_\theta$ and $D_\pi$

also share the same set of layers before their respective output layer. The input to scAAnet includes $\bar{x}_i$ and $l_i$, where $l_i$ is library size of cell $i$ calculated by the total number of counts and $\bar{x}_i$ is the expression profile of cell $i$ normalized by the library size. In the above formulation, $a_i$ and $b_i$ are two latent vectors generated from the encoders. Here, $a_i$ is a usage vector that is used as the input for decoders to estimate the three parameters of the ZINB distribution, while $b_i$ is an auxiliary vector that helps constrain the distance between archetypes and data points in the latent space. Moreover, we set positions for archetypes in the latent space in matrix $Z^{fix}$ and by multiplying $a_i$ with $Z^{fix}$, we can get locations of data points relative to the preset archetypes ($e_i$). This information is used for constructing an archetypal constraint in the loss function.

The activation functions chosen for the layers that output mean, dispersion, and dropout probability are Softmax, softplus, and sigmoid, respectively. Softmax transformation is applied to the outputs from encoders and a leaky ReLU activation is used for other hidden layers.

To increase flexibility, scAAnet allows users to choose from Poisson, ZI-Poisson, and NB distributions apart from ZINB distributions. In addition, users are able to choose whether the dispersion parameter is both cell and gene specific or constant across cells for the same gene.

<u>Construction of loss function</u>: The loss function of scAAnet consists of two parts, a reconstruction error term and an archetypal constraint. Since we model count scRNA-seq data using ZINB distributions, the reconstruction error is quantified by the negative log-likelihood (NLL) of ZINB distributions using the three parameters estimated by scAAnet:

$$RE = \sum_{i=1}^{n} NLL_{ZINB}(x_i; \mu_i l_i, \theta_i, \pi_i) \tag{2}$$

where $x_i$ is the unnormalized raw count data. The archetypal constraint of scAAnet is represented as the Frobenius norm between two matrices:

$$\iota_{AT} = \left\| Z^{fix} - B^T E \right\|_F^2 \tag{3}$$

In this term, we multiply the auxiliary matrix $B$ (the $i$th column is $b_i$) with the computed locations of data points relative to $Z^{fix}$ ($E$, whose $i$th row is $e_i$) to get the inferred locations of archetypes. Then, we indirectly constrain the distance between archetypes and the data in the latent space by penalizing the distance between preset archetypes and inferred archetypes.

<u>Algorithm</u>: We train scAAnet using the following algorithm:

**Algorithm 1** Training of scAAnet

**Input**: raw count scRNA-seq data $x_i$, $i = 1,\ldots,n$; number of archetypes $K$; maximum epoch number $E$; batch size $b$; number of warm-ups $W$.

**Output**: usage $a_i$, $i = 1,\ldots,n$; archetypes $z_k$, $k = 1,\ldots,K$; reconstructed expression profile $\hat{x}_i$, $i = 1,\ldots,n$.

**Init**: initialize network parameters using Xavier normal initializer and $Z^{fix}$

```
for m = 1 to E do # iterate through E epochs
for q = 1 to ceiling(n/b) do # iterate through mini-batches
 sample a mini-batch of size b
 calculate loss L_q = RE_q + l_{AT_q}
 if m≤W then
   update network parameters through backpropagation using the
RMSprop optimizer
 else
   update network parameters and Z^{fix} through backpropagation using
the RMSprop optimizer
 end if
end for
if the early stopping criterion is satisfied then
 break
```

```
end for
for i = 1 to n do
```
$a_i = f_{E_a}(\bar{x}_i)$
$\hat{x}_i = f_{D_\mu}(a_i Z^{fix})l_i$
```
end for
for k = 1 to K do
```
$z_k = f_{D_\mu}(Z_k^{fix})$
```
end for
```

The initial learning rate used by scAAnet is 0.01 and we reduce the learning rate by a factor of 0.5 if the loss does not improve in a given number of epochs (by default, 10). The minimum learning rate can be reduced to is 0.001. We also adopt an early stopping approach, in which training is stopped if the loss does not improve in a given number of epochs (set to 100) and the model weights are restored from the best epoch.

Although the locations of the archetypes in the latent space are set in the beginning, our algorithm allows $Z^{fix}$ to be trainable after a warm-up period to make the shape of latent space more flexible to the input data. We found that scAAnet achieved a good balance between the accuracy of cell usage inference and the accuracy of GEP inference on simulated data after a warm-up period of 20 (S14 Fig). After 20, the performance was relatively robust as the number of warm-up periods increased. Therefore, we set the warm-up period to be 20 as the default value and adopt this strategy in real data analysis.

## Description and implementation details of methods

NMF [17]: NMF was implemented through function non_negative_factorization in the Python library scikit-learn (version 0.23.2). The expression profile of each cell was normalized by the total counts before performing NMF. Apart from the maximum iteration number, which was set as 400, all the other parameters were set as their default values.

AAnet [9]: AAnet (version 0.1.0) was downloaded from https://github.com/KrishnaswamyLab/AAnet. AAnet is an autoencoder neural network designed for non-linear archetypal analysis. AAnet forces the sum of loadings of data onto archetypes to be 1 by introducing a novel regularization on the middle layer, but the reconstruction error used by AAnet is still MSE. The expression count data need to be normalized before applying the AAnet model. According to the instruction of AAnet, we normalized data first by total counts and then applied a square root transformation. Parameters we chose for batch size, hidden layer size, and number of batches were 64, 128, and 10,000, respectively.

scVI [11]: scVI (version 0.9.0a2) was downloaded from https://github.com/YosefLab/scvi-tools. scVI is a deep generative model based on a variational autoencoder. The expression profile of each cell is modeled as a sample that follows a ZINB distribution and a latent embedding vector can be inferred by the encoder of scVI. The loss function of scVI is composed of two parts. The first part is the expectation of the logarithm of the conditional probability of the expression profile given the latent vector. The second part is the Kullback–Leibler divergence between a standard normal distribution and the inferred distribution of each latent vector. The latent vector is modeled as a normally distributed random vector in the traditional scVI, but it can be set to follow a logistic normal distribution to make it positive and sum to 1. In this case, scVI can be utilized to mimic the behavior of archetypal analysis. Since scVI models scRNA-seq data with a ZINB distribution, we do not need to perform any normalization on the raw count data. Parameters we used for batch size, hidden layer size, and the maximum epoch number were 64, 128, and 200, respectively.

LDVAE [18]: LDVAE was developed based on the framework of scVI and it is incorporated in the same Python package of scVI. Compared to scVI, LDVAE provides interpretability and

enables identification of gene programs by replacing the non-linear decoder with a linear reconstruction function, while keeping a non-linear inference model on the encoder side. Similar to scVI, we used raw count data without any normalization as the input for LDVAE and we used the same set of parameters as scVI.

## Simulation of synthetic single-cell RNA-seq data

We simulated synthetic scRNA-seq datasets using Splatter's [16] framework which is based on negative binomial distributions (equivalent to Gamma-Poisson hierarchical models). We reimplemented Splatter in Python tailored to our needs for archetypal analysis. For all simulated datasets, 4 GEPs were generated, which formed a 3-simplex with 4 vertices. The usage of these 4 GEPs for each cell was sampled from the simplex using a Dirichlet distribution. About 50% of the samples were generated from the boundary of the simplex by randomly setting one of the usages to be zero followed by normalization. The purpose of this construction is to allow the existence of cells that don't contain all the GEPs and to show scAAnet is able to handle such cells (see S1 Text section 8 for more discussion). Each dataset contained 2,000 genes and 3,000 cells. In each GEP, the proportion of DEGs among all the 2,000 genes was set to 10% and overlap of DEG sets among GEPs was allowed. The differential expression ratios for those genes were sampled from a log-normal distribution with the scale parameter as 1.0, and the location parameter as either 1.0, 0.75, or 0.5 to mimic different signal-to-noise ratios. For each signal-to-noise level, ten datasets were randomly generated. Other Splatter parameters were the same as those used in Kotliar et al. [3] and they were inferred by Splatter from 8,000 cells of the organoid dataset in Quadrato et al. [54]. Furthermore, we generated synthetic datasets that follow ZINB distribution by introducing a parameter $\lambda$ to control the dropout ratio. For cell $i$ and gene $j$, its dropout probability is defined as

$$\pi_{ij} = \exp(-\lambda \mu_{ij}^2) \tag{4}$$

where $\mu_{ij}$ is the mean expression level drawn from a Gamma distribution and such a relationship was used in a previous scRNA-seq study [55]. The values of $\lambda$ we chose were 0.3, 0.1, and 0.01. The dropout proportions in simulated datasets under different scenarios are shown in S15 Fig where Inf means NB distributions. With decreasing $\lambda$, the dropout proportion goes up.

## Single-cell RNA-seq data preprocessing

We used the Python library Scanpy [56] (version 1.7.2) to perform all the preprocessing and visualization of scRNA-seq data. Details for each dataset are provided below.

The pancreatic islet dataset [13]: This human scRNA-seq dataset contained 1,937 cells and 20,125 genes. We filtered out genes with non-zero expression in fewer than 3 cells and genes whose mean expression levels were lower than 0.001 across all cells before normalization. This dataset was prelabeled with a total of 14 cell types. We calculated the number of cells in each cell cluster and deleted cells whose cluster had fewer than 15 cells. One exception was for epsilon cells. There were only 13 of them but we decided to keep them because this cell type is an endocrine cell type along with other four cell types (alpha, beta, gamma, delta). The final dataset analyzed contained 1,908 cells and 14,739 genes. We then selected top 2,000 highly variable genes (HVGs) based on the pipeline in Scanpy for scAAnet. HVGs were selected by first modeling the relationship between mean and dispersion of all genes across cells. Genes with higher dispersion compared to genes with similar mean expression would be regarded as HVGs [57].

The lung IPF dataset [14]: The entire lung IPF data contained 312,928 cells which were classified into 38 discrete cell types and 45,947 unique transcripts. Here, we only used cells that

were annotated as fibroblast or myofibroblast because these two cell types were closely related so that they were suitable for archetypal analysis. We filtered out genes with non-zero expression in fewer than 10 cells and mean expression lower than 0.001. We also removed mitochondrial genes. The filtered dataset had 6,166 fibroblast and myofibroblast cells and 25,431 features. We then selected top 1,500 HVGs from this dataset for analysis according to the original paper.

The prefrontal cortex dataset [15]: We downloaded the filtered dataset provided by the paper, which contained 70,634 cells and 17,926 genes. We selected 1,920 microglial cells based on the provided cell label and top 2,000 HVGs for further analysis.

## UMAP visualization

We used UMAP [52,53] for the visualization of scRNA-seq datasets. Before applying UMAP, we ran a number of steps to prepare the data. First, cells were normalized by library size to a target sum of 10,000 counts. Then, they were log-transformed after adding one pseudo count. In the next step, the total counts per cell (and the percentage of mitochondrial genes for the lung dataset) were regressed out and the expression of each gene was scaled across cells with absolute values larger than 10 being clipped. Next, PCA was performed to reduce the dimensionality and top PCs were used to construct a neighborhood graph with a fixed size of local neighborhood. Last, the neighborhood graph was used to generate a two-dimensional UMAP. Details for each dataset are provided below.

The pancreatic islet dataset [13]: The number of top PCs we used for this dataset was 35 with default number of neighbors as 15. Apart from generating UMAP for the original expression profile using PCA, we also generated UMAPs based on the inferred usage matrix and on the reconstructed expression profile. For the former one, the neighborhood graph was constructed using the usage matrix instead of top PCs and the number of neighbors was 20. For the latter one, we repeated the previously described pipeline using the reconstructed matrix and the number of neighbors was also 20.

The lung IPF dataset [14]: After generating the UMAP based on the pipeline, we continued to conduct the following steps according to the original paper [14] of this dataset. The Louvain [58] clustering was run on the constructed neighborhood (resolution = 0.45). Then, partition-based graph abstraction [59] (PAGA) was used to quantify the connectivity among Louvain clusters and a diffusion map [60,61] was calculated using the top 5 PCs. Next, a new neighborhood graph was constructed with the neighborhood size as 80 using the diffusion map. Finally, a UMAP was generated on this new graph using PAGA as initial embedding positions (spread = 0.8).

The prefrontal cortex dataset [15]: We used top 10 PCs as in the original paper with default number of neighbors as 15 to generate the two-dimensional UMAP coordinates.

## Selection of number of archetypes (K)

Note that there is no universally definitive way to select the number of archetypes (K), but we still use some criteria to help the selection process. Generally speaking, we select K based on both the reconstruction error and stability of decomposition. For stability, we can assess both the stability of archetype inference and the stability of cell usage inference. To select K, we ran scAAnet 30 times using different random seeds for each candidate K. Candidate Ks are chosen from a reasonably wide range.

The reconstruction error is obtained from the first term in the loss function of scAAnet (Eq 2), which is the negative log-likelihood of a chosen count distribution. To evaluate the stability of cell usage inference under different numbers of archetypes, we use a similar method adopted from Brunet et al. [62]. For a cell usage inference result, we can assign each cell to a

dominant archetype based on its maximum cell usage. Across all 30 repeats, we can calculate the proportion that each pair of cells are assigned to the same archetype to get a consensus matrix C. The entries of C range from 0 to 1 and indicate the probability that two cells are assigned to the same archetype. If a K delivers stable cell usage results over 30 repeats, we expect entries of C are close to 0 or 1. To assess the stability of archetype inference under different numbers of archetypes, we borrow the idea from Kotliar et al. [3]. Under a given K, we concatenate 30 archetype matrices inferred using different random states. Then, we perform K-means clustering on the 30K archetypes and use Euclidean distance silhouette score to quantify the robustness of archetype inference across 30 repeats.

## Identification of top genes for archetypes

The first interesting question to be asked is what the dominating or top genes for each inferred GEP are. Here, we adopt a similar approach proposed by Gonzalez-Blas et al. [63] to identify top gene sets. After getting the archetype matrix Z from scAAnet, we can calculate gene scores across GEPs for each gene using the following equation:

$$S_{gk} = Z_{gk}\left(logZ_{gk} - \frac{\sum_{i=1}^{K} logZ_{gi}}{K}\right), \tag{5}$$

where $Z_{gk}$ is the scaled mean of gene $g$ in GEP $k$ and $K$ is the total number of archetypes. Then, we normalize the gene score to make it fall between 0 and 1:

$$NS_{gk} = \frac{S_{gk} - \min(S_{jk}, \forall j)}{\max(S_{jk}, \forall j) - \min(S_{jk}, \forall j)} \tag{6}$$

The next step is to identify the top gene set for each GEP based on normalized gene scores. A Gamma distribution can be fitted to the gene scores and a cutoff will be set based on a user-given probability threshold (default is 0.975). Any genes with normalized gene scores larger than the cutoff are collected into the top gene set for that GEP.

## Differentially expressed gene (DEG) test for archetypes

We designed a test for identifying archetype-specific DEGs. The test is based on marginal negative-binomial (NB) regression where we regress the count data of each gene $g$ onto the usage vector of each GEP $k$ across all the cells and using the log transformed total counts (library size) as the offset. The equation of our regression model is:

$$\log(E(x_{ig})) = \beta_{0gk} + \beta_{1gk}U_{ik} + \log l_i, i = 1, \dots, n.$$

where $x_{ig}$ is the raw count of gene $g$ in cell $i$, $U_{ik}$ is the inferred cell usage of GEP $k$ in cell $i$ and $l_i$ is the library size of cell $i$.

Multiple comparison adjustment like Bonferroni correction or Benjamini–Hochberg procedure can be performed after testing all the candidate genes. Two options of NB regression are provided in our test. The first one is the general NB regression with variance being $\mu + \frac{\mu^2}{\theta}$, where $\mu$ is the mean and $\theta$ is the dispersion parameter. The value of $\theta$ in the Python implementation of NB regression will be truncated if it's less than 0.5 or larger than 100. The second option is naïve NB regression where variance is equal to $\mu + \mu^2$. The second option is computationally more efficient without much loss of inference accuracy (S16 Fig).

To assess the performance of our DEG test using the continuous usage variable, we simulated datasets following similar steps as described in the first section in Methods. The only

difference was that instead of allowing different GEPs to have the same gene as their DEG, one gene could only appear as a DEG for one GEP. This simulation design was intentionally used to facilitate the assessment and model comparison.

### Gene set enrichment analysis

We performed gene set enrichment analysis for the upregulated DEGs identified in GEPs using hypergeometric tests. Gene sets including GO biological process [64,65] and Reactome [66] (version 7.4) were downloaded from the Molecular Signatures Database [67,68] (MSigDB). We tested for these two databases separately on each GEP and p values were adjusted by the Benjamini–Hochberg procedure. The cutoff for adjusted p-values was chose at 0.05.

## Supporting information

**S1 Fig. Selection of K in a simulated dataset.** (a) Stability of usage, (b) stability of archetypes and (c) reconstruction loss across different Ks for four count distributions. (d) Distance matrix showing archetype stability under K = 2–5. The darker the color the smaller the distance. (e) Consensus clustering matrices showing the usage stability under K = 2–5. The warmer the color the large the similarity.
(TIF)

**S2 Fig. Performance of scAAnet and other methods on synthetic scRNA-seq datasets in all settings.** (a) Results from datasets that were simulated under NB distributions. (b) Results under ZINB distributions with $\lambda = 0.3$. (c) Results under ZINB distributions with $\lambda = 0.1$. (d) Results under ZINB distributions with $\lambda = 0.01$. For each panel, figures from left to right are MSE between inferred cell usages and true usages of the 4 GEPs, Pearson correlation between inferred GEPs and true GEPs, and MSE between reconstructed scaled means and true means. Each box and whisker plot was plotted based on ten simulated datasets. Central lines represent medians, boxes represent the IQR, and the upper/lower whisker represents the largest/smallest value no further than $1.5 \times$ IQR. D.E.: differential expression.
(TIF)

**S3 Fig. MDS interpolation visualization of archetypal space recovered from scAAnet.** (a) scAAnet with MSE, (b) scAAnet with Poisson, (c) scAAnet with ZIP, and (d) scAAnet with NB distributions as the reconstruction error term. Red dots are the locations of archetypes. Figures in each column are colored by the true cell usage of the corresponding archetype (GEP).
(TIF)

**S4 Fig. Comparison of DEG test results between using continuous usages and discrete group assignment on synthetic scRNA-seq datasets.** (a) Datasets were simulated under NB, (b) ZINB with $\lambda = 0.3$, (c) ZINB with $\lambda = 0.1$, and (d) ZINB with $\lambda = 0.01$. For each panel, figures from left to right are TPR at q-value < 0.05 and FPR at q-value < 0.05 calculated across different signal-to-noise ratio levels. Bonferroni correction was used to obtain q-values. Each box and whisker plot was plotted based on ten simulated datasets. Central lines represent medians, boxes represent the IQR, and the upper/lower whisker represents the largest/smallest value no further than $1.5 \times$ IQR. TPR: true positive rate; FPR: false positive rate.
(TIF)

**S5 Fig. Selection of K in the pancreatic islet dataset.** (a) Stability of usage, (b) stability of archetypes and (c) reconstruction loss across different Ks for four count distributions. (d)

Consensus clustering matrices showing the usage stability under K = 8–12. The warmer the color the large the similarity.
(TIF)

**S6 Fig. MDS visualization of archetypal space in the pancreatic islet dataset.** (a) MDS visualization colored by cell types. Locations of inferred archetypes are shown as black dots. (b) MDS visualization colored by the inferred usage of the ten GEPs. Locations of inferred archetypes are shown as red dots.
(TIF)

**S7 Fig. REACTOME enrichment results of the two GEPs related to beta cells in the pancreatic islet dataset.** (a) Top enriched terms identified using significantly upregulated genes in GEP 2. (b) Top 10 enriched terms identified using significantly upregulated genes in GEP 7.
(TIF)

**S8 Fig. scAAnet is able to identify two GEPs using beta cells only that are similar to GEP 2 and GEP 7 identified in the pancreatic islet dataset.** (a) The UMAP visualization of the pancreatic islet scRNA-seq dataset, with beta cells highlighted in red. Black dots are locations of cells that have the largest usage of the 2 GEPs (marked in Arabic numerals). (b) UMAPs colored by inferred cell usage for each GEP. (c) Heatmap showing the Pearson correlations among GEP 1 and GEP 2 identified using beta cells only and GEP 2 and GEP 7 identified using the entire dataset. (d) The top 10 enriched GO biological process terms using 35 significantly upregulated genes of GEP 1. (e) The top 10 enriched GO biological process terms using 130 significantly upregulated genes of GEP 2.
(TIF)

**S9 Fig. Selection of K in the lung fibroblast and myofibroblast cells.** (a) Stability of usage, (b) stability of archetypes and (c) reconstruction loss across different Ks for four count distributions. (d) Consensus clustering matrices showing the usage stability under K = 2–5. The warmer the color the large the similarity.
(TIF)

**S10 Fig. MDS visualization of archetypal space in the lung fibroblast and myofibroblast cells.** (a) MDS visualization colored by cell types. (b) MDS visualization colored by disease groups. Locations of inferred archetypes in a and b are shown as black dots. (c) MDS visualization colored by the inferred usage of the three GEPs. Locations of inferred archetypes are shown as red dots.
(TIF)

**S11 Fig. REACTOME enrichment results of the three GEPs in the lung fibroblast and myofibroblast cells.** (a) Top enriched terms significantly upregulated genes in GEP 1, (b) GEP 2 and (c) GEP 3, respectively.
(TIF)

**S12 Fig. MDS visualization of archetypal space of microglial cells in the prefrontal cortex dataset.** (a) MDS visualization colored by subclusters. (b) MDS visualization colored by AD pathology groups. (c) MDS visualization colored by sexes. Locations of inferred archetypes in a, b and c are shown as black dots. (d) MDS visualization colored by the inferred usage of the four GEPs. Locations of inferred archetypes are shown as red dots.
(TIF)

**S13 Fig. Sex and AD group differences in cell usage of 4 GEPs in each Microglia subcluster.** Box and whisker plot of the usage of each GEP in cells of subcluster Mic0, Mic1, Mic2, and

Mic3 (from top to bottom), colored by (a) AD pathology group and by (b) sex, respectively. Central lines represent medians, boxes represent the IQR, and whiskers represent the 5th and 95th quantiles.
(TIF)

**S14 Fig. Performance of scAAnet over increasing number of warm-up.** (a) Cell usage inference performance. MSE between inferred usage and true usage was calculated. (b) GEP inference performance. Pearson correlations between inferred GEPs and true GEPs were calculated. For both a and b, red dashed lines are vertical lines plotted at the number of warm-up periods being 20. Ticks along x-axis are numbers of warm-up periods (0, 5, 10, 20, 30, 40, 50, 70, 100, 150, and 200). The total number of epochs was fixed at 200 in this experiment, so if the number of warm-up period is 200, it means $Z^{fix}$ is not allowed to be trainable.
(TIF)

**S15 Fig. Dropout proportions of synthetic scRNA-seq datasets across different values of $\lambda$ and mean D.E. log2 fold change of GEP-specific genes.** Each box and whisker plot was plotted based on ten simulated datasets. Central lines represent medians, boxes represent the interquartile range (IQR), and the upper/lower whisker represents the largest/smallest value no further than $1.5 \times$ IQR. D.E.: differential expression.
(TIF)

**S16 Fig. Comparison of DEG test results among different regression models on synthetic scRNA-seq datasets.** (a) Datasets were simulated under NB, (b) ZINB with $\lambda = 0.3$, (c) ZINB with $\lambda = 0.1$, and (d) ZINB with $\lambda = 0.01$. For each panel, figures from left to right are TPR at q-value $< 0.05$ and FPR at q-value $< 0.05$ calculated across different signal-to-noise ratio levels. Bonferroni correction was used to obtain q-values. Each box and whisker plot was plotted based on ten simulated datasets. Central lines represent medians, boxes represent the IQR, and the upper/lower whisker represents the largest/smallest value no further than $1.5 \times$ IQR. TPR: true positive rate; FPR: false positive rate; AUC: area under curve.
(TIF)

**S17 Fig. Influence of relative weight of the archetypal loss on scAAnet.** (a) Change of the archetypal loss. (b) Change of the reconstruction loss. (c) Change of the total loss. (d) Performance on cell usage inference across different weights. (e) Performance on GEP inference across different weights. (f) Performance on reconstruction accuracy across different weights.
(TIF)

**S18 Fig. Robustness of scAAnet to misspecified K in a simulated dataset.** (a) Correlation heatmap between pairs of archetypes inferred using K = 3 and K = 4 (left), and between inferred using K = 5 and K = 4 (right). (b) UMAPs colored by the inferred GEP usage. From top to bottom are results based on K being 3, 4, and 5, respectively. Orange arrows indicate the progression of GEPs as K increases.
(TIF)

**S19 Fig. Robustness of scAAnet to misspecified K across all simulation settings.** Each bar represents a simulation setting where ten repeated experiments were run under different seeds. The percentage on the y-axis was calculated as the number of experiments where GEPs inferred under K = 3 constituted a perfect subset of GEPs inferred under K = 4 divided by ten. A perfect subset was defined as no overlaps among the GEPs inferred under K = 4 that had the largest correlations with the GEPs inferred under K = 3.
(TIF)

**S20 Fig. Performance of scAAnet and other methods on datasets simulated with no shared GEPs.** (a) Results from datasets that were simulated under NB distributions. (b) Results under ZINB distributions with $\lambda = 0.3$. (c) Results under ZINB distributions with $\lambda = 0.1$. (d) Results under ZINB distributions with $\lambda = 0.01$. Other details are the same as S2 Fig.
(TIF)

**S21 Fig. scAAnet identified GEPs corresponding to milestones in simulated trajectory-based data.** (a) Linear trajectory with three milestones colored by inferred GEP usage. (b) Bifurcation trajectory with four milestones colored by inferred GEP usage. Milestones are marked as black dots and backbone of the trajectory is shown in black curves.
(TIF)

**S22 Fig. UMAP visualization using NMF results on the pancreatic islet dataset.** From left to right are UMAPs plotted using the top 35 PCs of the observed scRNA-seq data, using the inferred cell usage matrix, and using the reconstructed expression matrix. UMAPs are colored by 10 known cell clusters. Black dots are locations of cells that have the largest usage of the corresponding GEP (marked in Arabic numerals).
(TIF)

**S23 Fig. UMAP visualization using NMF results on microglial cells in the prefrontal cortex dataset.** (a) UMAP colored by microglia subclusters. Black dots are locations of cells that have the largest usage of the corresponding GEP (marked in Arabic numerals). (b) UMAPs colored by the inferred usage of each GEP.
(TIF)

**S24 Fig. The level of sharing of GEPs in the real data.** Barplots showing the distribution of the largest inferred usage across cells grouped by annotated clusters in the pancreas islet dataset (a), the lung fibroblasts and myofibroblasts (b), and the microglial cells (c).
(TIF)

**S25 Fig. The level of sparsity of inferred usage in the real data.** Barplots showing the distribution of the smallest inferred usage across cells grouped by annotated clusters in the pancreas islet dataset (a), the lung fibroblasts and myofibroblasts (b), and the microglial cells (c).
(TIF)

**S26 Fig. Heatmaps of normalized gene scores of the top 20 DEGs across different GEPs.** (a) In a simulated dataset, where top is from the true GEP matrix and bottom is from the inferred the GEP matrix (rearranged GEPs to match the order of true GEPs by Pearson correlation); (b) in the pancreatic islet dataset (where the bottom if a subset of the top figure without IAPP for a better resolution); (c) in the lung dataset; (d) in the prefrontal cortex dataset.
(TIF)

**S1 Table. Top genes identified in each GEP in the pancreatic islet dataset.**
(XLSX)

**S2 Table. Upregulated DEGs (adjusted p-value < 0.05) of GEP 2 in the pancreatic islet dataset.**
(XLSX)

**S3 Table. Upregulated DEGs (adjusted p-value < 0.05) of GEP 7 in the pancreatic islet dataset.**
(XLSX)

**S4 Table. Enriched GO terms of GEP 2 in the pancreatic islet dataset.**
(XLSX)

**S5 Table. Enriched GO terms of GEP 7 in the pancreatic islet dataset.**
(XLSX)

**S6 Table. Enriched Reactome terms of GEP 2 in the pancreatic islet dataset.**
(XLSX)

**S7 Table. Enriched Reactome terms of GEP 7 in the pancreatic islet dataset.**
(XLSX)

**S8 Table. Upregulated DEGs (adjusted p-value < 0.05) of GEP 1 identified only using beta cells in the pancreatic islet dataset.**
(XLSX)

**S9 Table. Upregulated DEGs (adjusted p-value < 0.05) of GEP 2 identified only using beta cells in the pancreatic islet dataset.**
(XLSX)

**S10 Table. Top genes identified in each GEP in lung fibroblasts and myofibroblasts.**
(XLSX)

**S11 Table. Upregulated DEGs (adjusted p-value < 0.05) of GEP 1 in lung fibroblasts and myofibroblasts.**
(XLSX)

**S12 Table. Upregulated DEGs (adjusted p-value < 0.05) of GEP 2 in lung fibroblasts and myofibroblasts.**
(XLSX)

**S13 Table. Upregulated DEGs (adjusted p-value < 0.05) of GEP 3 in lung fibroblasts and myofibroblasts.**
(XLSX)

**S14 Table. Enriched GO terms of GEP 1 in lung fibroblasts and myofibroblasts.**
(XLSX)

**S15 Table. Enriched GO terms of GEP 2 in lung fibroblasts and myofibroblasts.**
(XLSX)

**S16 Table. Enriched GO terms of GEP 3 in the lung fibroblasts and myofibroblasts.**
(XLSX)

**S17 Table. Enriched Reactome terms of GEP 1 in lung fibroblasts and myofibroblasts.**
(XLSX)

**S18 Table. Enriched Reactome terms of GEP 2 in lung fibroblasts and myofibroblasts.**
(XLSX)

**S19 Table. Enriched Reactome terms of GEP 3 in the lung fibroblasts and myofibroblasts.**
(XLSX)

**S20 Table. Upregulated DEGs (adjusted p-value < 0.05) of GEP 1 in the microglia of the prefrontal cortex dataset.**
(XLSX)

**S21 Table. Upregulated DEGs (adjusted p-value < 0.05) of GEP 2 in the microglia of the prefrontal cortex dataset.**
(XLSX)

**S22 Table. Upregulated DEGs (adjusted p-value $< 0.05$) of GEP 3 in the microglia of the prefrontal cortex dataset.**
(XLSX)

**S23 Table. Upregulated DEGs (adjusted p-value $< 0.05$) of GEP 4 in the microglia of the prefrontal cortex dataset.**
(XLSX)

**S24 Table. Enrichment of GEP-specific DEGs across sets of makers of the four microglia subclusters.** For each GEP, microglia subcluster with the lowest p-value is highlighted in red.
(XLSX)

**S25 Table. Enriched GO terms of GEP 2 in the microglia of the prefrontal cortex dataset.**
(XLSX)

**S26 Table. Enriched GO terms of GEP 3 in the microglia of the prefrontal cortex dataset.**
(XLSX)

**S27 Table. Enriched GO terms of GEP 4 in the microglia of the prefrontal cortex dataset.**
(XLSX)

**S1 Text. Supplementary notes for methods and additional results.**
(DOCX)

**S1 Data. Supplementary data of numerical values underlying main figures.**
(XLSX)

## Author Contributions

**Conceptualization:** Yuge Wang.

**Data curation:** Yuge Wang.

**Formal analysis:** Yuge Wang.

**Funding acquisition:** Hongyu Zhao.

**Investigation:** Yuge Wang.

**Methodology:** Yuge Wang.

**Project administration:** Yuge Wang.

**Resources:** Hongyu Zhao.

**Software:** Yuge Wang.

**Supervision:** Hongyu Zhao.

**Validation:** Yuge Wang.

**Visualization:** Yuge Wang.

**Writing – original draft:** Yuge Wang.

**Writing – review & editing:** Yuge Wang, Hongyu Zhao.

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
