## [Decision Letter · Decision Letter 0]

8 Dec 2021

Dear Prof. Zhao,

Thank you very much for submitting your manuscript "Non-linear archetypal analysis of single-cell RNA-seq data by deep autoencoders" for consideration at PLOS Computational Biology.

As with all papers reviewed by the journal, your manuscript was reviewed by members of the editorial board and by several independent reviewers. In light of the reviews (below this email), we would like to invite the resubmission of a significantly-revised version that takes into account the reviewers' comments.

We cannot make any decision about publication until we have seen the revised manuscript and your response to the reviewers' comments. Your revised manuscript is also likely to be sent to reviewers for further evaluation.

Sincerely,

Mingyao Li

Associate Editor

PLOS Computational Biology

Jian Ma

Deputy Editor

PLOS Computational Biology

Reviewer's Responses to Questions

**Comments to the Authors:**

Reviewer #1: Please see attached file.

Reviewer #2: In this manuscript, Wang and Zhao propose scAAnet, a deep autoencoder for non-linear archetypal analysis of single-cell RNA-seq data. Building on the assumption that the expression profile of each cell results from a non-linear combination of multiple gene expression programs (GEPs), scAAnet aims to detect GEPs and infer the relative activity of each GEP across cells. There are many merits in this work. The manuscript is well organized and clearly written. The methods are innovative and clearly presented, based on conceptually reasonable assumptions. The simulations were carefully designed and novel biological insights revealed from applications to several real datasets.

I have two relatively major comments. First, scAAnet assumes that cell types share some GEPs. While this is conceptually justifiable, can the authors quantify the level of shared GEPs in real data, or evaluate how would scAAnet perform if the cells don’t share GEPs in the extreme case? Second, scAAnet needs to preset archetype. It is unclear how to preset the values. How would scAAnet’s performance vary with different hyperparameter choices. In the pseudo codes, there seems a warm up period where the Z^fix will be updated simultaneously. However, if that is the case, it seems that Eq ([Disp-formula pcbi.1010025.e014]) on page 32 is no longer identifiable?

I have some additional minor comments.

1. Can GEPs be used to infer cell types? Would it be possible to use the inferred mixture of GEPs to infer cell type?

2. GEPs identified in real data show their values in revealing underlying latent factors, pathways or network. I assume other scRNA-seq methods can also be used for similar purposes, in more convoluted manners, for example, by first clustering the cells and then performing analysis based on cell type differentially expressed genes? Presumably, without the meaningful reduction into GEPs, alternative methods that do not summarize data into GEPs would provide much noisier conclusions? The authors can consider trying competing methods on some of their real datasets.

3. The authors showed that shared pattern across cell types/disease can be obtained from inferred GEPs. But we can also analyze the cell type marker genes/ disease marker genes and compare the similarity of marker genes in different cell types/ diseases. What will be the benefit of scAAnet if the GEP analysis eventually goes back to GEP marker gene analysis?

Reviewer #3: The paper proposes an interesting model for understanding the gene expression programs underlying cells from a population with continuous cell states. The nonnegativity and convexity required in archetypal decomposition enable ease of interpretation. Compared with linear archetypal decomposition by NMF, scAAnet incorporates nonlinearity naturally through an autoencoder.

- The paper has discussed several conceptual advantages of scAAnet including nonlinearity and the inclusion of the archetypal constraint. While it is clear the method achieves better results on the simulated data, I was wondering if some comparison can be performed on any of the real data examples, in particular with NMF based methods. Does the nonlinearity allow scAAnet to capture different biological signals? Some comparison will help practitioners better understand the utility of the method, given scAAnet comes with additional computational cost.

- The paper has explained how to choose K, the number of archetypes, through stability criteria. How robust is the method to actually misspecified K? It is a point worth investigating at least on simulated data.

- The archetypal constraint assumes all cells lie in a tight convex hull in the latent space, which seems a bit restrictive to me. For example, many developmental data have trajectories with bifurcation structure. It would be great if the authors could provide more discussion on the feasibility of this assumption.

- Related to the question above, one could imagine increasing K or using a larger convex hull would accommodate for more general shapes. In this sense, is there some trade-off involved in the use of archetypal constraint loss? Usually when different losses are combined, they are weighed differently with tuning parameters. Does the relative weight of reconstruction loss vs. archetypal loss affect any part of the results?

- It is stated that in simulation 50% of the samples were generated from the boundary of the simplex, which seems a bit contrived. What is the purpose of this construction? Would it be more realistic to include some outliers outside this simplex?

**Have the authors made all data and (if applicable) computational code underlying the findings in their manuscript fully available?**

Reviewer #1: **No: **The method is on GitHub but "Analysis codes for generating results in this paper are available upon request by email." It would be good if these are published as well.

Reviewer #2: Yes

Reviewer #3: Yes

PLOS authors have the option to publish the peer review history of their article (what does this mean?). If published, this will include your full peer review and any attached files.

Reviewer #1: No

Reviewer #2: No

Reviewer #3: No
---

## [Decision Letter · Decision Letter 1]

4 Mar 2022

Dear Prof. Zhao,

Thank you very much for submitting your manuscript "Non-linear archetypal analysis of single-cell RNA-seq data by deep autoencoders" for consideration at PLOS Computational Biology. As with all papers reviewed by the journal, your manuscript was reviewed by members of the editorial board and by several independent reviewers. The reviewers appreciated the attention to an important topic. Based on the reviews, we are likely to accept this manuscript for publication, providing that you modify the manuscript according to the review recommendations.

Sincerely,

Mingyao Li

Associate Editor

PLOS Computational Biology

Jian Ma

Deputy Editor

PLOS Computational Biology

[LINK]

Reviewer's Responses to Questions

**Comments to the Authors:**

Reviewer #1: The authors have addressed most of my comments. I would like to see a few supplementary figures added to improve the interpretation of the GEPs.

Specifically, regarding the inferred GEP matrix. The GEPs corresponds to the archetypes, which, as the authors describe, represent “pure types or extrema of the real data”.

In the pancreatic data set, most cell types are primarily associated with one GEP, to the extent that we might even annotate that GEP using the cell type name. Epsilon cells are associated with GEP 3 (gamma) and GEP 5(delta), and the authors provide the interpretation that Epsilon cells are similar to both gamma and delta cells.

How does one interpret that the beta cells having very high usage of either GEP2 or GEP7? The authors compared GEP2 and GEP7 and concluded that these contain different top genes, and the genes are associated with different GO terms. Does this suggest that there are two different subtypes of beta cells, one subtype resembling archetype 2 and another resembling archetype7?

The authors provide in Fig 10 the top 20 DEGs for GEP2 and GEP 7. But lacking an overlap between the top 20 genes does not directly reflect how much GEP2 and GEP7 differ. Can the authors provide a heatmap for the GEP matrix (as described on Line 178-179)? The usage of the GEPs are visualized (for example, in Fig 5c). There are 2000 HVGs from 1937 cells, so the GEP matrix is not much larger than the GEP usage matrix. A heatmap can provide a rather straightforward view of the GEPs, and help the reader interpret the GEPs, in addition to the top DEG and GO analysis that the authors already provide.

It would be useful to add these in supplementary figures:

• For the simulated data, heatmaps of the true GEP matrix and the inferred GEP matrix(after re-arranging the GEPs to best match the true order)

• For the real data sets, heatmaps of the inferred GEP matrix

Minor:

The DEG test for archetypes

Line 799 : is $m_i$ the library size? To be consistent with the notation used earlier, maybe it’s better to use $l_i$ again.

In the results for the lung fibroblast and myofibroblast cells: “GEP 3 is the one not only cell-type-specific but also disease-related.” However, Fig 7 seems to suggest that in Myofibroblasts, both GEP2 and GEP3 usage are disease related?

Reviewer #2: The authors have carefully addressed all my comments. I have no additional suggestions.

Reviewer #3: The authors have addressed all my comments.

**Have the authors made all data and (if applicable) computational code underlying the findings in their manuscript fully available?**

Reviewer #1: Yes

Reviewer #2: Yes

Reviewer #3: Yes

PLOS authors have the option to publish the peer review history of their article (what does this mean?). If published, this will include your full peer review and any attached files.

Reviewer #1: No

Reviewer #2: No

Reviewer #3: No

Figure Files:

Data Requirements:

Reproducibility:

References:

---

## [Editor Report · Decision Letter 2]

15 Mar 2022

Dear Prof. Zhao,

We are pleased to inform you that your manuscript 'Non-linear archetypal analysis of single-cell RNA-seq data by deep autoencoders' has been provisionally accepted for publication in PLOS Computational Biology.

Best regards,

Mingyao Li

Associate Editor

PLOS Computational Biology

Jian Ma

Deputy Editor

PLOS Computational Biology

---

## [Editor Report · Acceptance letter]

25 Mar 2022

PCOMPBIOL-D-21-01867R2 

Non-linear archetypal analysis of single-cell RNA-seq data by deep autoencoders

Dear Dr Zhao,

I am pleased to inform you that your manuscript has been formally accepted for publication in PLOS Computational Biology. Your manuscript is now with our production department and you will be notified of the publication date in due course.

With kind regards,

Zsofia Freund
